# Molecular motion and tridimensional nanoscale localization of kindlin control integrin activation in focal adhesions

Thomas Orré[1], Adrien Joly[1,5], Zeynep Karatas[1,5], Birgit Kastberger[2], Clément Cabriel [3], Ralph T. Böttcher [4], Sandrine Lévêque-Fort [3], Jean-Baptiste Sibarita [1], Reinhard Fässler [4], Bernhard Wehrle-Haller [2], Olivier Rossier [1,6✉] & Grégory Giannone [1,6✉]

Focal adhesions (FAs) initiate chemical and mechanical signals involved in cell polarity, migration, proliferation and differentiation. Super-resolution microscopy revealed that FAs are organized at the nanoscale into functional layers from the lower plasma membrane to the upper actin cytoskeleton. Yet, how FAs proteins are guided into specific nano-layers to promote interaction with given targets is unknown. Using single protein tracking, super-resolution microscopy and functional assays, we link the molecular behavior and 3D nanoscale localization of kindlin with its function in integrin activation inside FAs. We show that immobilization of integrins in FAs depends on interaction with kindlin. Unlike talin, kindlin displays free diffusion along the plasma membrane outside and inside FAs. We demonstrate that the kindlin Pleckstrin Homology domain promotes membrane diffusion and localization to the membrane-proximal integrin nano-layer, necessary for kindlin enrichment and function in FAs. Using kindlin-deficient cells, we show that kindlin membrane localization and diffusion are crucial for integrin activation, cell spreading and FAs formation. Thus, kindlin uses a different route than talin to reach and activate integrins, providing a possible molecular basis for their complementarity during integrin activation.

[1] University Bordeaux, CNRS, Interdisciplinary Institute for Neuroscience, IINS, UMR, Bordeaux, France. [2] Department of Cell Physiology and Metabolism, Centre Médical Universitaire, Geneva 4, Switzerland. [3] Institut des Sciences Moléculaires d'Orsay, CNRS UMR8214, Univ. Paris-Sud, Université Paris Saclay, Orsay, Cedex, France. [4] Max Planck Institute of Biochemistry, Martinsried, Germany. [5] These authors contributed equally: Adrien Joly, Zeynep Karatas. [6] These authors jointly supervised this work: Olivier Rossier, Grégory Giannone. ✉email: olivier.rossier@u-bordeaux.fr; gregory.giannone@u-bordeaux.fr

Cell adhesion, polarity and migration rely on the assembly of integrin-based adhesive structures called focal adhesions (FAs). At the core of FAs, integrins establish physical connections between the extracellular matrix (ECM) and the actin cytoskeleton[1] by recruiting directly and indirectly hundreds of different proteins with diverse structural, signaling or scaffolding functions, collectively called the integrin adhesome[2,3]. In addition to this elaborate network of protein interactions, 3D super-resolution microscopy revealed that FAs are composed of multiple functional nano-layers located at different distances from the plasma membrane and composed of specific adhesome components: namely a membrane-proximal integrin signaling layer, a force transduction layer and an upper actin regulatory layer[4,5]. Partitioning of FAs components into nano-domains in the plane of the plasma membrane[6,7] as well as in the axial direction[4] spatially restricts the combination of possible molecular interactions and thereby focuses protein functions to specific locations. For instance, inactive vinculin is recruited to the integrin layer, whereas activated vinculin is localized in the upper layers to mediate connection to the actin cytoskeleton[8]. The next challenge is now to characterize the precise sequence of molecular events guiding proteins to specific nano-layers to foster interactions with specific protein targets. To establish the link between molecular motions and functions of proteins, it is necessary to study the molecular path that proteins follow in the complex 3D cellular environment.

This is particularly true for the regulators of integrin activation, a central molecular process occurring inside integrin-based adhesions and controlling integrin transitions between "inactive" unbound and "active" ECM-bound states[9]. This process is reversible and involves numerous activators and inhibitors that compete for adjacent and overlapping binding sites on the short cytoplasmic tail of αβ-integrins[10,11]. Talin, the most described integrin activator, is recruited inside FAs directly from the cytosol[6] and controls integrin immobilization by binding to the proximal NPxY motif on integrin β cytoplasmic tail with its FERM domain (4.1-protein/ezrin/radixin/moesin)[10]. More recently, kindlins were demonstrated to be critical for integrin activation[12–14]. Kindlins also possess a FERM domain which binds to β-integrin via the distal NxxY motif[15]. Integrin activation by kindlins has been predominantly described during cell spreading and the formation of nascent adhesions (NAs) within active protrusions of the cell[16–18] rather than in mature FAs. Nevertheless, talins and kindlins were demonstrated to cooperate during integrin activation[11,18,19]. Indeed, kindlin is often required for proper integrin activation by talin[15,18,20–25]. Moreover, kindlin and talin activate integrin not simply in an additive manner, but rather in a synergistic way[15,18,25,26]. Thus, kindlins and talins may cooperate to activate integrins by binding to distinct regions of the β integrin tail; however, the molecular events underlying this cooperation are poorly understood[19].

The majority of knowledge about integrin activation is paradoxically derived from studies in which the complexity of the adhesive structures found in adherent cells was reduced or even absent. For instance, integrin activation is mainly assessed in suspended cells by flow cytometry using soluble ligand[12,26]. In the same line, structural studies or biochemistry including immunoprecipitations and integrin tail peptide pull-downs are often performed in vitro with purified proteins or truncated protein domains[27–29]. Contrasting with the molecular intricacy found in cells, these approaches are essential to determine critical domains in protein-protein interactions, but remain inadequate to detect transient interactions occurring within cells. In addition, these approaches fail to provide the duration and subcellular location of molecular interactions underlying important cellular functions.

The recent development of single protein localization and tracking (SPT) techniques has provided the possibility to quantitatively study protein motions in their native environment, where all mechanisms of regulation are ongoing[6,30–33]. In the case of FA proteins, it was possible to correlate integrin activation with its immobilization inside FAs[6,34] and elucidate how membrane nanotopology controls integrin activation[35]. SPT also revealed the path used by talin to reach integrins[6]. Thus SPT techniques are complementary to biochemical approaches and have become instrumental to establish protein dynamics as a readout of protein functions in complex macromolecular assemblies.

Here, we directly link the molecular behavior of kindlin-2 with its cellular function, i.e., integrin activation, by combining SPT, super-resolution microscopy and functional rescue assays in genetically engineered cells. Albeit being a cytosolic protein, we show that kindlin resides and undergoes free diffusion along the plasma membrane to reach integrins, in sharp contrast to talin, which enters FAs directly from the cytosol. Using 3D super-resolution microscopy, we show that kindlin-2 resides in the membrane-proximal integrin layer. Deletion of kindlin-2 Pleckstrin Homology (PH) domain (kindlin-2-ΔPH) dramatically decreased kindlin membrane recruitment and diffusion, accumulation in FAs and integrin-mediated cell adhesion. Importantly, the fraction of kindlin-2-ΔPH that accumulated inside FAs was located above the membrane-proximal integrin layer. Restoring kindlin-2-ΔPH membrane diffusion and localization in the membrane-proximal integrin layer with a nonspecific lipid anchor was sufficient to restore recruitment to FAs and integrin-mediated cell adhesion. These results show that an efficient interaction of kindlin-2 with integrins inside FAs cannot be triggered by cytosolic diffusion alone, but rather requires kindlin-2 membrane diffusion to drive kindlin-2 to the proper functional layer. Thus, we show that kindlin-2 uses a different route from talin to reach integrins, providing a possible molecular basis for their complementarity during integrin activation. This study demonstrates that the path used by a protein to reach its target is a critical cellular determinant of its function.

## Results

**Kindlin controls integrin immobilization and free diffusion cycles within mature FAs.** Using super-resolution microscopy and SPT, we demonstrated that integrins reside in FAs through free diffusion and immobilization cycles, during which activation promotes immobilization[6,36]. Here we used the same strategy to explore whether kindlin is also involved in cycles of integrin activation within mature FAs (Fig. 1). For this purpose, mouse embryonic fibroblasts (MEFs) were seeded on fibronectin (FN)-coated glass coverslips. MEFs were co-transfected with mEos2-fused proteins and GFP-paxillin as a FA reporter. We performed high-frequency sptPALM acquisition (50 Hz) and sorted trajectories occurring outside and inside FAs. Trajectories were sorted according to their diffusion mode (free diffusion, confined diffusion, immobilization; Fig. 1a–d, g, h), and diffusion coefficients (D) were computed (Fig. 1e, f; Methods). As we showed before, both β1-WT-mEos2 and β3-WT-mEos2 (Fig. 1a, c) displayed a large fraction of immobilization inside FAs but also a significant fraction of membrane free diffusion (Fig. 1g, h). Outside FAs, the fraction and rate of free diffusion increased at the expense of immobilizations (Fig. 1e–h and Supplementary Fig. 1g, h). Importantly, a mutation in the membrane-distal NxxY motif (β1-Y795A or β3-Y759A), which decreases interaction with kindlin[37], decreased the fraction of immobilized β1-Y795A-mEos2 and to a smaller extent of β3-Y759A-mEos2 (Fig. 1b, d, e–h). These mutations also increased the rate of free diffusion inside FAs (Supplementary Fig. 1g, h). We

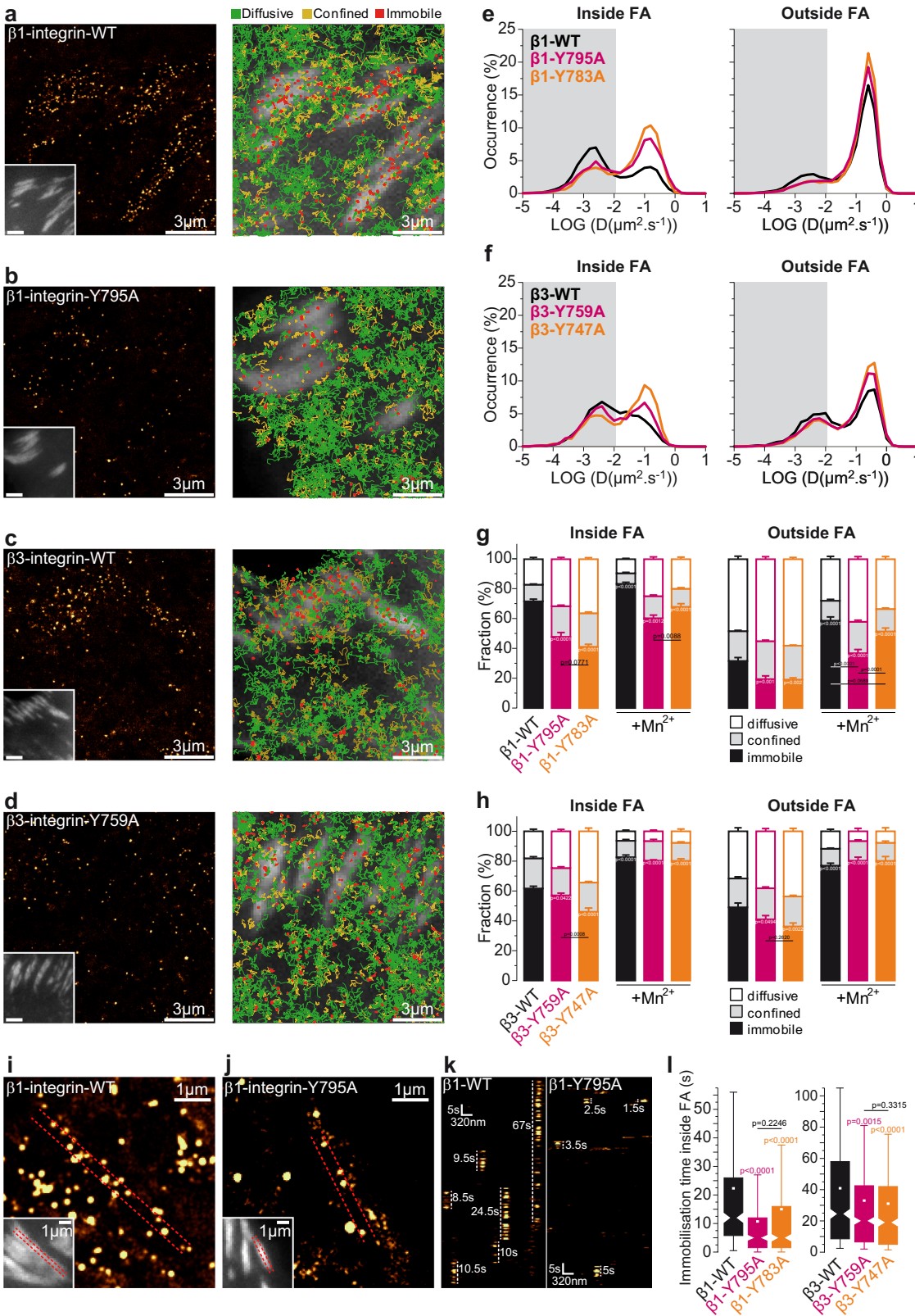

generated super-resolved time-lapses and kymographs (Fig. 1i–k) to compare β-integrin immobilization times and found that immobilization durations were longer for β1-WT-mEos2 and β3-WT-mEos2 compared to the kindlin-binding defective mutant β1-Y795A-mEos2 and β3-Y759A-mEos2 (Fig. 1k, l). Integrin activation by kindlin was shown to be critical for the formation of NAs and membrane protrusions[17,18]. As a consequence, kindlin-deficient cells do not form mature FAs, thus preventing to study integrin activation by kindlin in those structures. Our results indicate that kindlin controls integrin immobilization inside mature FAs, suggesting that kindlin also plays a crucial role during integrin activation in mature FAs.

Consistent with our previous results[6], a mutation in the membrane-proximal NPxY (β1-Y783A[38] or β3-Y747A[39]), which

**Fig. 1 Kindlin is required for β-integrin immobilization.** Left: Super-resolution PALM intensity images of β1-WT-mEos2 (**a**), β1-Y795A-mEos2 (**b**), β3-WT-mEos2 (**c**), β3-Y759A-mEos2 (**d**) in MEFs obtained from sptPALM sequences (50 Hz, >80 s). Inset: low resolution image of GFP-paxillin, which was co-expressed for FAs labeling (scale bar: 3 μm). Right: color-coded trajectories overlaid on FAs labeled by GFP-paxillin (greyscale) show the diffusion modes: free diffusion (green), confined diffusion (yellow) and immobilization (red). **e** Distributions of the diffusion coefficient D computed from the trajectories of β1-WT-mEos2 (black), β1-Y795A-mEos2 (pink), β1-Y783A-mEos2 (orange) obtained inside (left) and outside FAs (right), are shown in a logarithmic scale. The gray area including D values inferior to $0.011 \, \mu m^2.s^{-1}$ corresponds to immobilized proteins. Values represent the average of the distributions obtained from different cells. **f** Same as **e**, but with β3-WT-mEos2 (black), β3-Y759A-mEos2 (pink), β3-Y747A-mEos2 (orange). **g** Fraction of β1-WT and mutants undergoing free diffusion, confined diffusion or immobilization inside (left) and outside FAs (right) with or without manganese activation ($Mn^{2+}$). Values represent the average of the fractions obtained from different cells (error bars: SEM). **h** Same as **g**, but for β3-WT and mutants. Super-resolution PALM intensity image of β1-WT (**i**) and β1-Y795A (**j**) density in a FA (inset: labeled with GFP–paxillin) obtained from a sptPALM sequence. **k** Kymographs generated from a sptPALM time-lapse image along the length of a FA as shown by the dashed lines encompassing the FA in **i** and **j**. **l** Box plots displaying the median (notch) and mean (square) ± percentile (box: 25–75%, whiskers: 10–90%) of immobilization time inside FAs of β1-WT and mutants (left) and of β3-WT and mutants (right). **a–h**: results for β1-WT-mEos2 without (17 cells) and with $Mn^{2+}$ (15 cells), β1-Y795A-mEos2 without (22 cells) and with $Mn^{2+}$ (22 cells), β1-Y783A-mEos2 without (16 cells) and with $Mn^{2+}$ (20 cells), β3-mEos2 without (16 cells) and with $Mn^{2+}$ (13 cells), β3-Y759A-mEos2 without (20 cells) and with $Mn^{2+}$ (11 cells, 2 ind. exp.) and β3-Y747A-mEos2 without (16 cells) and with $Mn^{2+}$ (10 cells, 2 ind. exp.) correspond to pooled data from three independent experiments unless indicated. For (**l**), 6 (β1-Y795A, β1-Y783A, β3-WT), 5 (β3-Y759A, β3-Y747A), or 3 cells (β1-WT) corresponding to pooled data from three independent experiments were analysed. Where indicated, statistical significance was obtained using two-tailed, non-parametric Mann–Whitney rank sum test. Inside and outside FAs, the different conditions without $Mn^{2+}$ were compared to the corresponding β-integrin-WT condition; with $Mn^{2+}$, each given condition was compared with the value obtained without $Mn^{2+}$. Otherwise, a black line indicates which conditions were compared. The exact P values are indicated on the figure except when $P < 0.0001$. Source data are provided as a Source Data file.

was shown to decrease the interaction with talin, filamin and tensin[39], also decreased the fraction and duration of immobilization for β1-783A-mEos2 and β3-Y747A-mEos2 (Fig. 1g, h, l). Importantly, whereas β3-integrins showed a stronger immobilization defect with a mutation in the talin- over the kindlin-binding site, immobilization of β1-integrins was decreased similarly by both mutations. Thus, our data show that kindlin binding appears as critical as talin binding for β1-integrin immobilization. Furthermore, integrin activation by $Mn^{2+}$ triggered massive integrin immobilization both inside and outside FAs for β3-WT[6], but also for β3-747A (talin mutant) and β3-759A (kindlin mutant) (Fig. 1h and Supplementary Fig. 1c, d, f). However, $Mn^{2+}$ treatment did not increase immobilization of kindlin-mutant β1-795A as efficiently as for β1-WT or talin-mutant β1-783A (Fig. 1g and Supplementary Fig. 1a, b, e). This finding reveals that, contrary to β3-integrins, $Mn^{2+}$ cannot bypass the requirement of kindlin to fully activate β1-integrins. This could explain why in the presence of kindlin but not talin, cells could spread after $Mn^{2+}$ treatment or initiate binding to FN and resist force[18], while in the presence of talin but not kindlin, cells treated with $Mn^{2+}$ were unable to initiate spreading or binding to FN[18]. Together, these findings show that binding to kindlin is more critical for β1-integrin than for β3-integrin immobilization in mature FAs.

**Kindlin-2 undergoes free diffusion along the plasma membrane, inside and outside FAs.** Talin is recruited to FAs directly from the cytosol without prior membrane free diffusion, nor association with freely-diffusive integrins outside FAs[6]. The three kindlin isoforms (kindlin-1, −2, −3) possess structural similarities with the talin FERM domain but differ in their FERM subdomain F2 with the presence of a PH domain[40,41]. To decipher how kindlin is recruited to FAs and reaches integrins, we performed sptPALM with the ubiquitously expressed isoform, kindlin-2 (mEos2-kindlin-2-WT; Fig. 2a). As for talin-1 and both β-integrin isoforms, kindlin-2 was essentially immobile and enriched in FAs (Fig. 2a, d). Interestingly, using super-resolved time-lapses and kymographs (Fig. 2f–h) we found that immobilization durations were much shorter for kindlin-2 compared to talin-1 (Fig. 2i). Analysis of longer kymographs showed that the fraction of immobilized kindlin-2 undergoing retrograde flow with speed above $2 \, nm.s^{-1}$ was around 40%, similar to what was measured for talin-1 (Fig. 2j, k)[6]. Our results suggest that the interaction of kindlin-2 with stationary

integrins is more prevalent than with F-actin, because the fraction of F-actin moving rearward above $2 \, nm.s^{-1}$ is around 75%[6]. Importantly and unlike talin-1 (Fig. 2b, Supplementary Movie S1), kindlin-2 displayed membrane free diffusion both inside and outside FAs (Fig. 2a, c–e, Supplementary Movie S1). We obtained the same results for kindlin-1 (Supplementary Fig. 2). Furthermore, as kindlin was shown to be important for the formation of NAs[16–18], we also studied the diffusive behavior of kindlin-2 in NAs within the lamellipodia of spreading MEFs[31]. Like found in mature FAs, mEos2-kindlin-2 is immobile and enriched in NAs (Supplementary Fig. 3) and displays membrane free diffusion both inside and outside NAs (Supplementary Fig. 3). Altogether, these results indicate that membrane free diffusion is a general feature of the kindlin family that could confer distinct capabilities compared to talin during integrin activation in NAs and FAs.

**Disruption of the integrin-kindlin interaction increases kindlin diffusion inside FAs.** We then investigated the molecular basis of kindlin-2 membrane free diffusion. The diffusive behavior of kindlin-2-WT was unaffected by $Mn^{2+}$ treatment, demonstrating that kindlin-2 membrane diffusion is not influenced by the membrane diffusion of β-integrins (Fig. 3a, c–e). This is consistent with the slower rate of free diffusion of kindlin-2 compared to integrins (Fig. 2e), observed both inside and outside FAs. Kindlin-2 could therefore reach integrins inside FAs using membrane free diffusion. Then, association between integrin and kindlin, which contributes to integrin immobilization (Fig. 1), could reciprocally lead to kindlin-2 immobilization. To test this hypothesis, we used point mutations QW614/615AA on kindlin-2 that decrease binding to β-integrins (kindlin-2-QW) according to previous biochemistry assays[14,20,23,25,26,42]. Kindlin-2-QW displayed a decreased immobile fraction compared to kindlin-2-WT (Fig. 3b, d), mirroring the effects observed with the β1-795A and β3-Y759A mutations (Fig. 1b, d, g, h). These results suggest that inside FAs, kindlin population consists of a freely diffusing pool detached from integrins and an immobile pool engaged with integrins. We obtained the same results for kindlin-2-QW in NAs (Supplementary Fig. 3). The remaining fraction of immobilized kindlin-2-QW could result from binding to other partners, especially paxillin[17], ILK[20] or from residual binding of kindlin-2-QW to integrin. However, a point mutation decreasing kindlin-2 binding to ILK (kindlin-2-L357A)[20] had no effect on kindlin-2 immobilizations in FAs (Supplementary Fig. 4).

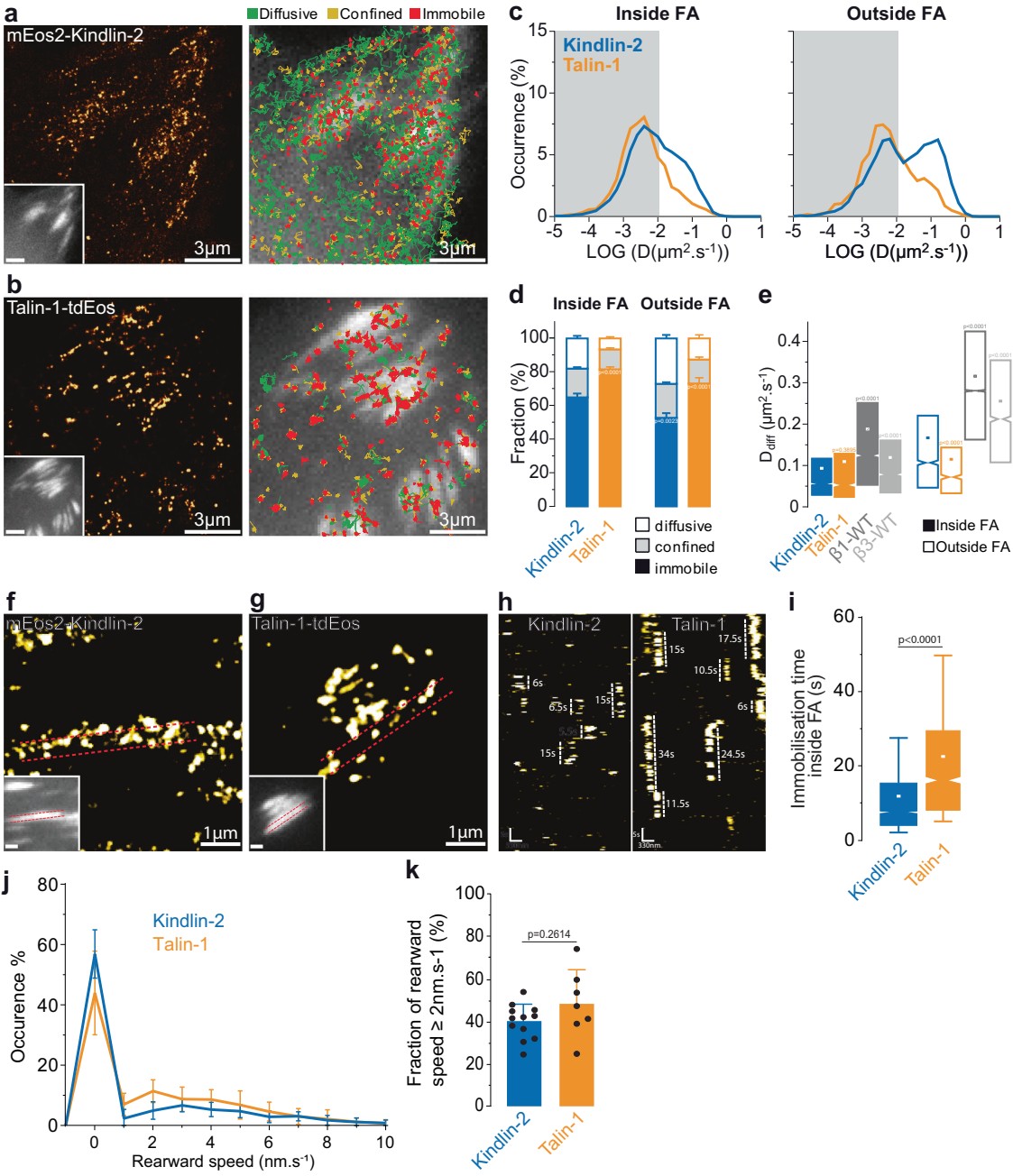

These immobilizations could result from ILK-independent recruitment of kindlin-2 inside FAs, as described previously in ILK deficient fibroblasts[20], potentially by binding to integrins, paxillin[18], or other unknown partners. Alternatively, transfected kindlin-2-L357A mutants can outcompete endogenous kindlin-2 in FAs, as showed in kindlin-2 knock down fibroblasts[20]. However, in our experimental conditions, the level of mEos2-Kindlin-2-L357A expression was only 1.5-fold the one of endogenous kindlin-2 (Supplementary Fig. 4), which indicates that in our MEFs ILK-independent recruitment of kindlin is predominant. Thus, our results support the idea that integrins are playing a crucial role in kindlin immobilization inside mature FAs.

**Kindlin-2 PH domain drives membrane recruitment and free diffusion.** Kindlin-2 could use multiple phospholipid binding motifs found on its F0, F1, and PH domains to associate with the plasma membrane[12,23,40,43]. A point mutation (K390A) that abolished kindlin-2 PH domain interaction with phosphoinositides in vitro[23,40] only slightly reduced membrane free diffusion of kindlin-2-K390A outside FAs (Supplementary Fig. 4). On the other hand, deletion of the entire PH domain (kindlin-2-ΔPH) strongly inhibited membrane free diffusion, both inside and outside FAs (Fig. 4a, c, d, Supplementary Movie S2). This finding suggests that kindlin-2 membrane free diffusion is mainly mediated by its PH domain. Consistent with this hypothesis, the kindlin-2 PH domain alone is able to diffuse freely along the plasma membrane (Fig. 4b–e). Like talin, paxillin-mEos2 is recruited and immobilized in FAs directly from the cytosol, since we detected almost no membrane free diffusion both inside and outside FAs (Supplementary Fig. 5). This shows that kindlin free diffusion does not result from co-diffusion with paxillin, which kindlin can bind to via its F0 and PH domains[17,18]. To evaluate recruitment of kindlin-2 to the plasma membrane, we compared

**Fig. 2 Kindlin-2 undergoes lateral free diffusion along the plasma membrane.** Left: Super-resolution PALM intensity images of mEos2-kindlin-2 (**a**) and talin-1-tdEos (**b**) in MEFs obtained from a sptPALM sequence (50 Hz, >80 s). Inset: low resolution image of GFP-paxillin, which was co-expressed for FAs labeling (scale bar: 3 μm). Right: color-coded trajectories overlaid on FAs labeled by GFP-paxillin (greyscale) show the diffusion modes: free diffusion (green), confined diffusion (yellow) and immobilization (red). **c** Distributions of the diffusion coefficient D computed from the trajectories of talin-1-tdEos (yellow) and mEos2-kindlin-2 (blue) obtained inside (left) and outside FAs (right), are shown in a logarithmic scale. The gray area including D values inferior to 0.011 μm².s⁻¹ corresponds to immobilized proteins. Values represent the average of the distributions obtained from different cells. **d** Fraction of mEos2-kindlin-2 (blue) and talin-1-tdEos (yellow) undergoing free diffusion, confined diffusion or immobilization inside (left) and outside FAs (right). Values represent the average of the fractions obtained from different cells (error bars: SEM). **e** Box plots displaying the median (notch) and mean (square) ± percentile (25–75%) of diffusion coefficients corresponding to the free diffusion trajectories inside (left) and outside FAs (right). Super-resolution PALM intensity image of mEos2-kindlin-2 (**f**) and talin-1-tdEos (**g**) density in a FA (inset: labeled with GFP–paxillin) obtained from a sptPALM sequence.
**h** Kymographs generated from a sptPALM time-lapse image along the length of a FA as shown by the dashed lines encompassing the FA in **f** and **g**. **i** Box plots displaying the median (notch) and mean (square) ± percentile (box: 25–75%, whiskers: 10–90%) of immobilization time inside FAs for kindlin-2 and talin-1. **j** Distribution of rearward speed (mean ± s.d. for cells) for kindlin-2 and talin-1. **k** Fraction of rearward speed equal or superior to 2 nm.s⁻¹ (mean ± s. d. for cells) for kindlin-2 and talin-1. Corresponding data points are overlaid on the bar chart as dot plots. Results for mEos2-kindlin-2 (**a–e**: 13 cells; **f–k**: 1127 immobilization events in 12 cells) correspond to pooled data from three independent experiments. Results for β1-WT-mEos2 (dark gray) and β3-WT-mEos2 (light gray) corresponds to data shown in Fig. 1 and Supplementary Fig. 1. Results for talin-1-tdEos (**a–e**: 8 cells; **f–k**: 584 immobilization events in 7 cells) correspond to pooled data from two independent experiments. Where indicated, statistical significance was obtained using two-tailed, non-parametric Mann–Whitney rank sum test. Inside and outside FAs, the different conditions were compared to the corresponding mEos2-kindlin-2 condition Otherwise, a black line indicates which conditions were compared. The exact P values are indicated on the figure except when P < 0.0001. Source data are provided as a Source Data file.

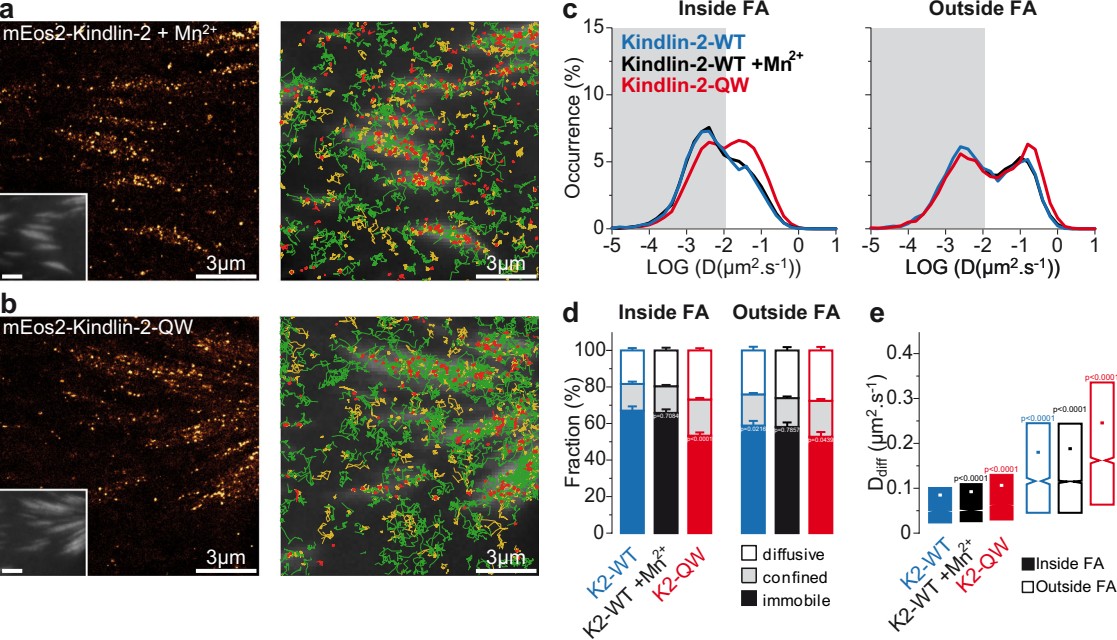

**Fig. 3 Kindlin-2 interaction with integrin induces kindlin-2 immobilization, but is dispensable for kindlin-2 diffusion.** Left: Super-resolution PALM intensity images of mEos2-kindlin-2-WT in Mn²⁺-stimulated MEFs (**a**) and mEos2-kindlin-2-QW614/615AA in MEFs (**b**) obtained from a sptPALM sequence (50 Hz, >80 s). Inset: low resolution image of GFP-paxillin, which was co-expressed for FAs labeling (scale bar: 3 μm). Right: color-coded trajectories overlaid on FAs labeled by GFP-paxillin (greyscale) show the diffusion modes: free diffusion (green), confined diffusion (yellow) and immobilization (red). **c** Distributions of the diffusion coefficient D computed from the trajectories of mEos2-kindlin-2-WT (blue), mEos2-kindlin-2-WT in Mn²⁺-stimulated MEFs (black), mEos2-kindlin-2-QW614/615AA (red) obtained inside (left) and outside FAs (right), are shown in a logarithmic scale. The gray area including D values inferior to 0.011 μm².s⁻¹ corresponds to immobilized proteins. Values represent the average of the distributions obtained from different cells. **d** Fraction of proteins undergoing free diffusion, confined diffusion or immobilization inside (left) and outside FAs (right) for mEos2-kindlin-2-WT, mEos2-kindlin-2-WT in Mn²⁺-stimulated MEFs, mEos2-kindlin-2-QW614/615AA (same color-code). Values represent the average of the fractions obtained from several cells (error bars: SEM). **e** Box plots displaying the median (notch) and mean (square) ± percentile (25–75%) of diffusion coefficients corresponding to the free diffusion trajectories inside (left) and outside FAs (right). Results for mEos2-kindlin-2-WT (17 cells) and mEos2-kindlin-2- QW614/615AA (33 cells) correspond to pooled data from five independent experiments. Results for mEos2-kindlin-2-WT + integrin activation by Mn²⁺ (17 cells) correspond to pooled data from four independent experiments. Where indicated, statistical significance was obtained using two-tailed, non-parametric Mann–Whitney rank sum test. Inside and outside FAs, the different conditions were compared to the corresponding mEos2-kindlin-2-WT condition. The exact P values are indicated on the figure except when P < 0.0001. Source data are provided as a Source Data file.

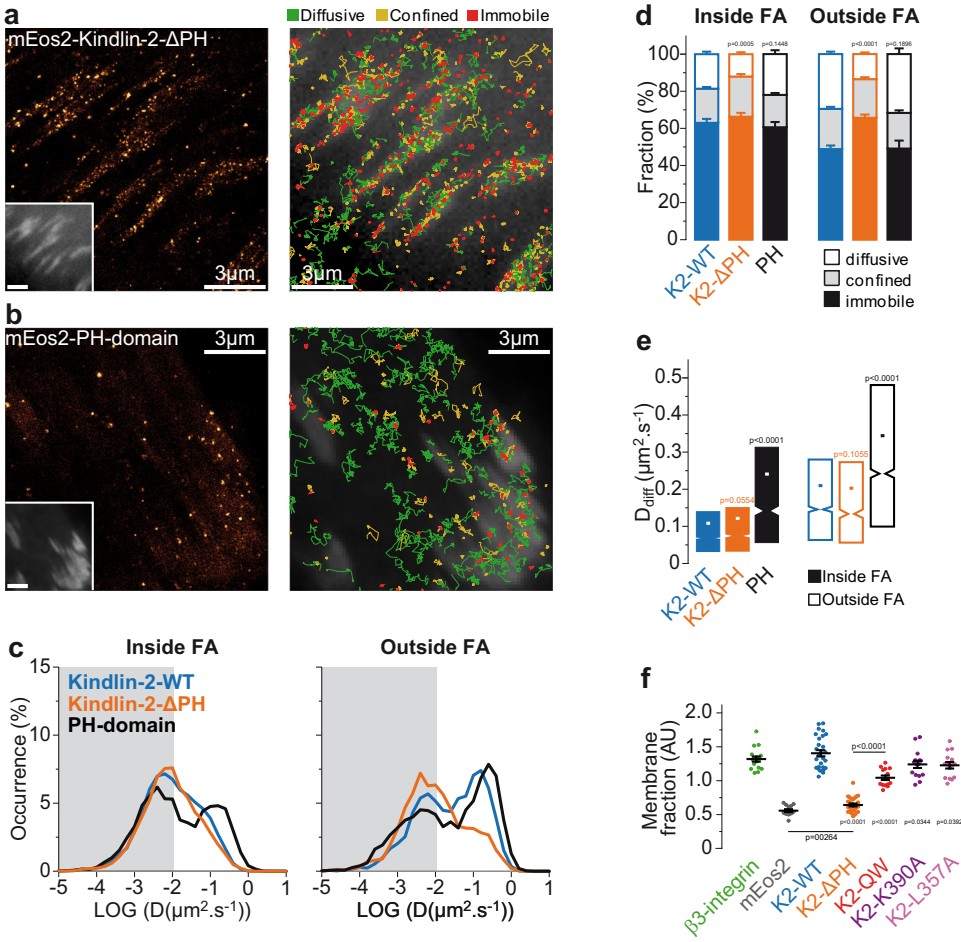

**Fig. 4 Kindlin-2 membrane recruitment and diffusion depends on the pleckstrin homology domain.** Left: Super-resolution PALM intensity images of mEos2-kindlin-2-ΔPH (**a**) and mEos2-PH-domain of kindlin-2-WT (**b**) in MEFs obtained from a sptPALM sequence (50 Hz, >80 s). Inset: low resolution image of GFP-paxillin, which was co-expressed for FAs labeling (scale bar: 3 μm). Right: color-coded trajectories overlaid on FAs labeled by GFP-paxillin (greyscale) show the diffusion modes: free diffusion (green), confined diffusion (yellow) and immobilization (red). **c** Distributions of the diffusion coefficient D computed from the trajectories of mEos2-kindlin-2-WT (blue), mEos2-kindlin-2-ΔPH (orange) and mEos2-PH-domain of kindlin-2-WT (black) obtained inside (left) and outside FAs (right), are shown in a logarithmic scale. The gray area including D values inferior to 0.011 μm².s⁻¹ corresponds to immobilized proteins. Values represent the average of the distributions obtained from different cells. **d** Fraction of proteins undergoing free diffusion, confined diffusion or immobilization inside (left) and outside FAs (right) for mEos2-kindlin-2-WT, mEos2-kindlin-2-ΔPH, and mEos2-PH-domain of kindlin-2-WT (same color-code). Values represent the average of the fractions obtained from different cells (error bars: SEM). **e** Box plots displaying the median (notch) and mean (square) ± percentile (25–75%) of diffusion coefficients corresponding to the free diffusion trajectories inside (left) and outside FAs (right). Results of sptPALM (in **a**–**e**) for mEos2-kindlin-2-WT (15 cells), mEos2-kindlin-2-ΔPH (28 cells) and mEos2-PH-domain of kindlin-2-WT (18 cells) correspond to pooled data from three independent experiments. **f** Fraction of proteins recruited at the membrane defined as the ratio of the membrane-level fluorescence signal (TIRF) to the total fluorescence signal of the cell (epifluorescence). Results of membrane fraction (in **f**) for mEos2-kindlin-2-WT (28 cells), mEos2-kindlin-2-ΔPH (27 cells) correspond to pooled data from two independent experiments, whereas results for β3-integrin-mEos2 (17 cells), cytosolic mEos2 (13 cells), mEos2-kindlin-2-QW (15 cells), mEos2-kindlin-2-K390A (15 cells) and mEos2-kindlin-2-L357A (15 cells) correspond to one experiment. Where indicated, statistical significance was obtained using two-tailed, non-parametric Mann–Whitney rank sum test. Inside and outside FAs, the different conditions were compared to the corresponding mEos2-kindlin-2-WT condition. Otherwise, a black line indicates which conditions were compared. The exact *P* values are indicated on the figure except when *P* < 0.0001. Source data are provided as a Source Data file.

the whole-cell fluorescence intensity of photoactivated mEos2 constructs to their membrane fluorescence, by using epifluorescence and TIRF imaging, respectively[44,45] (Fig. 4f, see "Methods"). To determine the lower and higher limits of membrane recruitment, we used respectively β3-integrin and cytosolic mEos2. Kindlin-2-WT displayed a high membrane recruitment that was almost abolished by deletion of the PH domain. On the contrary, kindlin-2-QW, kindlin-2-K390A and kindlin-2-L357A point mutations mildly decreased membrane recruitment, with only a minor effect for the K390A and L357A mutations. Thus, kindlin-2 membrane recruitment and diffusion, which could

favor kindlin interactions with integrins inside mature FAs, are mainly mediated by its PH domain.

**Kindlin-2 is acting in the integrin layer within mature FAs.** Kindlin-2 possesses multiple binding partners distributed in the different nano-layers of FAs, including integrins[14] and paxillin[18] in the membrane-proximal layer and actin[46] in the upper actin regulatory layer. Kindlin functions could rely on its relocalization from one layer to another, as seen for vinculin[8]. To test this hypothesis, we performed 3D super-resolution microscopy using direct optical nanoscopy with axially localized detection (DONALD)[47,48] (Fig. 5).

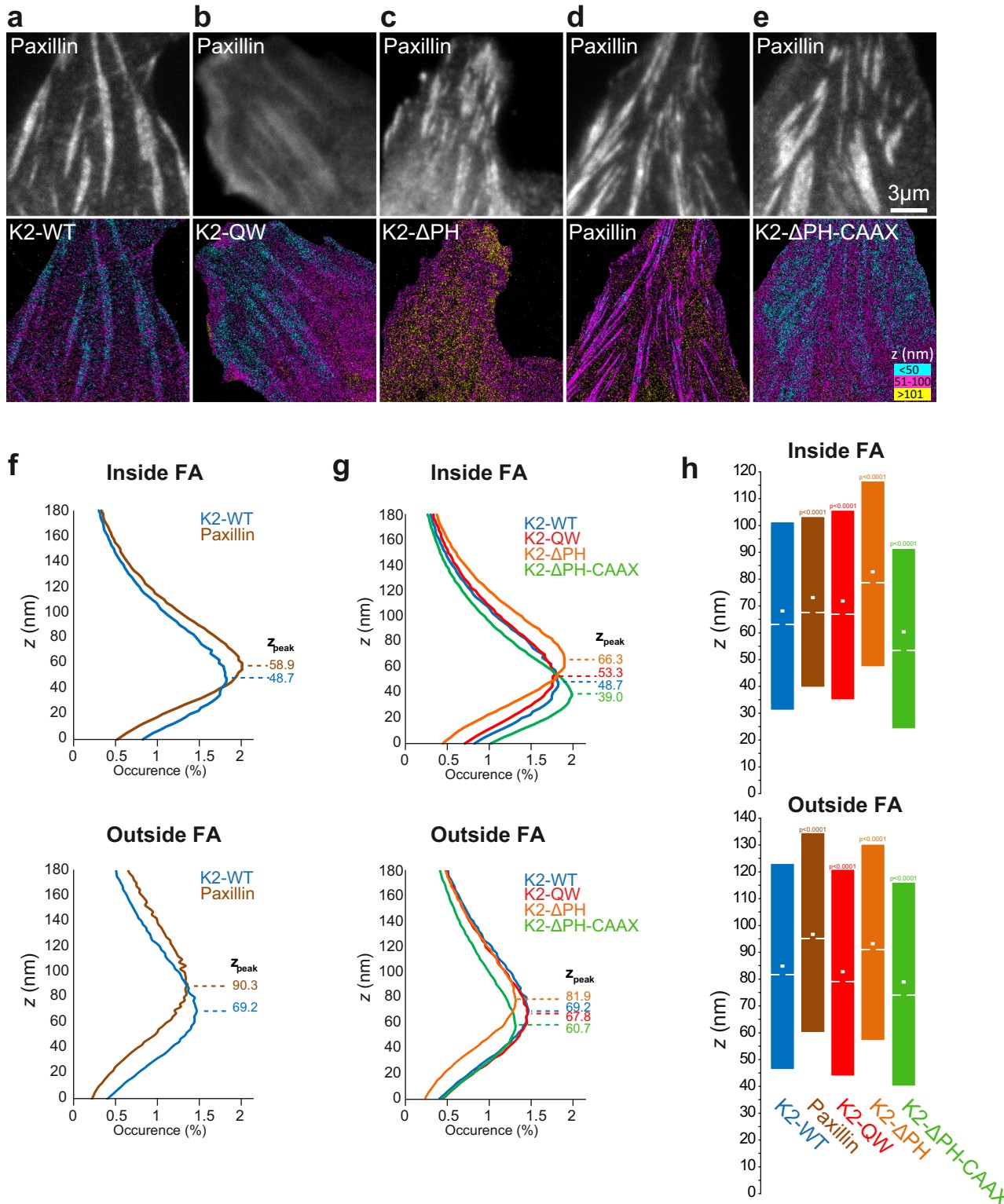

We used GFP-tagged proteins that we imaged by dSTORM using anti-GFP nanobodies labeled with AlexaFluor-647[49]. Like demonstrated previously[4], N-terminus labeled GFP-paxillin were located in the upper boundary of the membrane-proximal integrin layer ($z_{peak}$: 58.9 nm; Fig. 5d, f, h). Like paxillin, GFP-kindlin-2-WT was not widely distributed along the z axis of FAs, but concentrated in the integrin layer at the vicinity of the plasma membrane ($z_{peak}$: 48.7 nm; Fig. 5a, f–h). This shows that the strong kindlin-2 membrane recruitment (Fig. 4f) and free-diffusion (Fig. 2) we observed

occur in close proximity to integrins. While inhibiting kindlin binding to integrins (kindlin-2-QW) had a minor impact on kindlin localization ($z_{peak}$: 53.3 nm; Fig. 5b, g, h), deletion of the PH domain induced a large redistribution of kindlin-2 in the upper layers of FAs ($z_{peak}$: 66.3 nm; Fig. 5c, g, h). Altogether, these results confirm that kindlin membrane localization mainly depends on its PH domain and is not controlled by integrin binding. Moreover, these results show that the kindlin-2 main binding partners in mature FAs are localized inside the integrin layer, and that kindlin-2 is not

**Fig. 5 Kindlin localization in the integrin signaling layer inside FAs depends on the PH domain.** Top panel: Epifluorescence images of RFP-paxillin showing the localization of FAs. Bottom panel: 3D super-resolution images by DONALD displaying the axial localization of kindlin-2-WT (**a**), kindlin-2-QW (**b**), kindlin-2-ΔPH (**c**), paxillin (**d**) and kindlin-2-ΔPH-CAAX (**e**); scale bar, 3 μm. For each pixel, the average axial localization of detected single molecules is color-coded. **f** Distributions of the single molecule axial localizations of kindlin-2-WT (blue) and paxillin (brown) inside (top) and outside FAs (bottom). Values represent the average of the distributions obtained from different cells. Positions of occurrence maxima ($z_{peak}$) were estimated by Gaussian fitting. **g** Same as **f**, but with kindlin-2-WT (blue), kindlin-2-QW (red), kindlin-2-ΔPH (orange) and kindlin-2-ΔPH-CAAX (green). **h** Box plots displaying the median (notch) and mean (square) ± percentile (25–75%) of single molecule localizations inside (top) and outside FAs (bottom). Micrographs (**a**–**h**) correspond to a single experiment, but are representative of kindlin-2-WT: 6 cells, kindlin-2-QW: 9 cells, kindlin-2-ΔPH: 7 cells, paxillin: 6 cells, kindlin-2-ΔPH-CAAX: 7 cells. Where indicated, statistical significance was obtained using two-tailed, non-parametric Mann–Whitney rank sum test. The different conditions were compared to the mEos2-kindlin-2-WT condition. The exact $P$ values are indicated on the figure except when $P < 0.0001$. Source data are provided as a Source Data file.

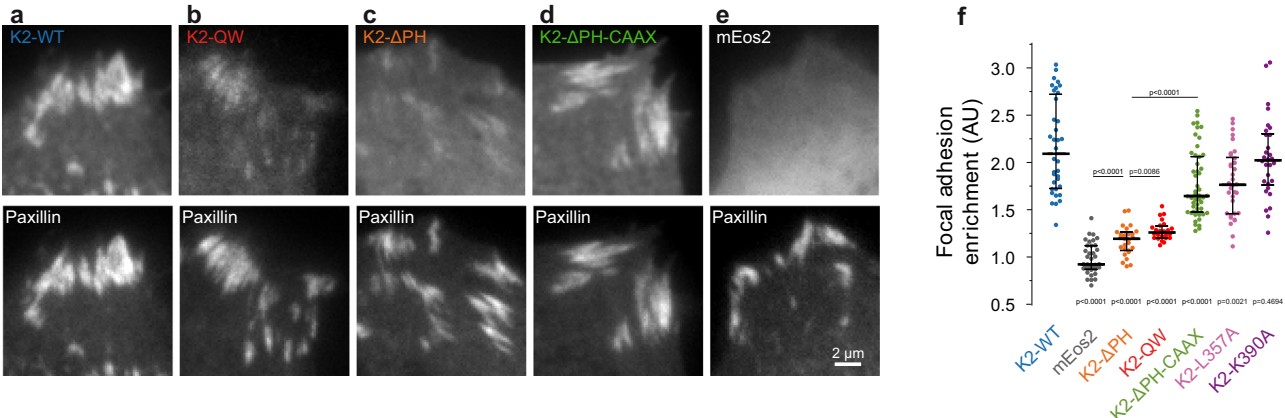

**Fig. 6 Kindlin 2 membrane recruitment and free diffusion is crucial for kindlin enrichment in FAs.** Top panel: TIRF images of photoconverted mEos2 coupled to kindlin-2-WT (**a**), kindlin-2-QW (**b**), kindlin-2-ΔPH (**c**), kindlin-2-ΔPH-CAAX (**d**), or photoconverted mEos2 alone (**e**). Bottom panel: TIRF images of GFP-paxillin showing the localization of FAs. Scale bar, 2 μm. **f** Enrichment inside FAs defined as the ratio of the average fluorescence intensity inside FAs to average fluorescence intensity outside FAs. Black bars represent medians and interquartile ranges. Displayed results correspond to pooled data from three independent experiments (kindlin-2-WT: 36 cells, kindlin-2-QW: 24 cells, kindlin-2-ΔPH: 27 cells, kindlin-2-ΔPH-CAAX: 50 cells, kindlin-2-L357A: 34 cells, kindlin-2-K390A: 30 cells, mEos2: 38 cells). Where indicated, statistical significance was obtained using two-tailed, non-parametric Mann–Whitney rank sum test. The different conditions were compared to the mEos2-kindlin-2-WT condition. Otherwise, a black line indicates which conditions were compared. The exact $P$ values are indicated on the figure except when $P < 0.0001$. Source data are provided as a Source Data file.

shifting between the integrin layer and other FA layers to fulfill its functions in FAs.

**Kindlin-2 membrane recruitment and free diffusion is crucial for kindlin enrichment in FAs.** Previous studies using conventional epifluorescence microscopy reported that kindlin-2-QW or kindlin-2-ΔPH could not be detected in FAs[12,14,20,23,42]. However, we observed residual accumulation in FAs of mEos2-kindlin-2-QW (Fig. 6b) and mEos2-kindlin-2-ΔPH (Fig. 6c) compared to cytosolic mEos2 (Fig. 6e), using TIRF microscopy. This is consistent with the immobilisations observed in FAs for these mutants by sptPALM (Fig. 3, Fig. 4). Importantly, we found that FA enrichment for kindlin-2-WT was higher than the sum of kindlin-2-ΔPH and kindlin-2-QW enrichments (Fig. 6f). This indicates that for kindlin-2-WT, interactions with integrins and those mediated by the PH domain have synergistic effects regarding accumulation in FAs. To test the importance during this process of membrane binding ensured by the PH domain, we restored membrane association of kindlin-2-ΔPH by adding a CAAX prenylation sequence. This induced membrane recruitment and diffusion of kindlin-2-ΔPH-CAAX inside and outside FAs (Supplementary Fig. 6a–c, e, Supplementary Movie S3). Furthermore, adding a CAAX motif also led to close association of kindlin-2-ΔPH to the membrane in the integrin layer ($z_{peak}$: 39 nm; Fig. 5e, g, h). Importantly, kindlin-2-ΔPH-CAAX enrichment in FAs was strongly increased compared to kindlin-2-ΔPH (Fig. 6c, d, f). Kindlin-2-QW, which diffuses normally along the plasma membrane but interacts weakly with integrins (Fig. 3), also

displayed a defective recruitment to FAs (Fig. 6b, f). Overall, our results demonstrate that two conditions must be fulfilled to enforce kindlin recruitment to FAs: (1) membrane recruitment and diffusion through the PH domain and (2) efficient binding to integrin.

**Kindlin-2 membrane recruitment and free diffusion is a key molecular event leading to integrin activation.** To show that kindlin-2 membrane recruitment and free diffusion are critical for kindlin function, i.e. integrin activation, we used kindlin-1 and kindlin-2 double deficient fibroblasts (Kind[Ko])[18] (Fig. 7). Kind[Ko] cells are unable to spread and form NAs and mature FAs, compared to Kind[Ctr] control cells, highlighting the central role of kindlin for integrin activation[15,18,26]. We observed that the diffusive behaviors of kindlin-2-WT, kindlin-2-QW, kindlin-2-L357A and kindlin-2-ΔPH in Kind[Ko] cells were similar to those observed in wild type MEFs expressing endogenous kindlin-1 and kindlin-2 (Supplementary Fig. 8), confirming that kindlin-2 membrane recruitment and diffusion are mostly driven by the PH domain (Supplementary Fig. 7e, 8). Furthermore, the similarity of diffusive behavior found in wild-type MEFs and Kind[Ko] cells suggests that sptPALM results obtained in WT MEFs are not biased by potential formation of heterodimers between endogenous kindlins and transfected kindlin-2 mutants as suggested by structural studies[28]. Using phase contrast microscopy, we assessed cell spreading and morphology on FN-coated coverslips 4 h after plating. Cells were partitioned into three categories: (i) non-spread; (ii) partially spread; (iii) fully spread (Fig. 7b). Kind[Ko] cells expressing paxillin-

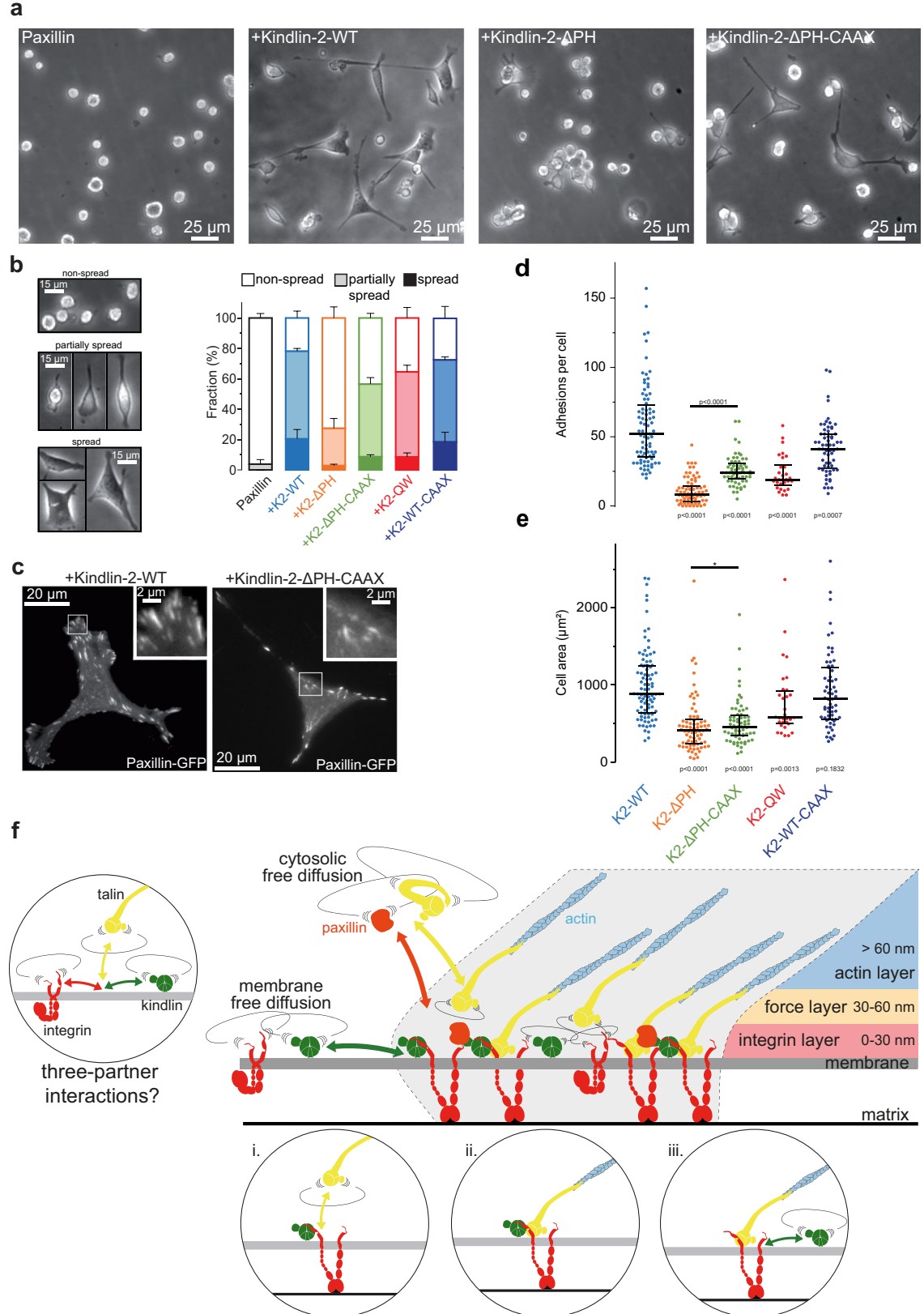

GFP were largely non-spread (Fig. 7a, b). Re-expression of kindlin-2-WT triggered cell spreading, although the majority of cells were partially spread with only 20 % displaying a fully spread morphology (Fig. 7a, b), and rescued integrin adhesion formation (Fig. 7c, d, e and Supplementary Fig. 9). We then tested the ability of kindlin-2 carrying point mutations/deletions to rescue cell

spreading and adhesive structure formation. This strategy enabled us to correlate the molecular behavior of modified kindlin-2 with their function as integrin activators. Expression of kindlin-2-QW, which displayed intermediate membrane recruitment, partially rescued spreading (Fig. 7b) and integrin adhesion formation (Fig. 7d, e and Supplementary Fig. 9). This suggests that kindlin-2-

**Fig. 7 PH-domain-mediated membrane recruitment and diffusion of kindlin-2 is crucial during cell spreading and mature FA formation. a** Phase contrast images showing cell spreading of Kind[Ko] (kindlin-1, kindlin-2 knock-out) cells after 4 h on fibronectin. Two days before the experiment, the cells were transfected to induce the expression of GFP-paxillin (leftmost), or GFP-paxillin along with mEos2-kindlin-2-WT (middle left), mEos2-kindlin-2-ΔPH (middle right), or mEos2-kindlin-2-ΔPH-CAAX (rightmost). Images representative of 3 independent experiments. **b** Top: Phase contrast images showing the morphological features of cells classified as non-spread, partially spread, or spread. Bottom: Relative fraction of non-spread, partially spread, and spread Kind[Ko] cells 2 days after re-expression of kindlin-2-WT or mutated variants and 4 h after seeding on fibronectin. Values represent the average of fractions from 3 independent experiments (error bars: SEM). Results for GFP-paxillin (141 cells), GFP-paxillin + mEos2-kindlin-2-WT (312 cells), GFP-paxillin + mEos2-kindlin-2-ΔPH (311 cells), GFP-paxillin + mEos2-kindlin-2-ΔPH-CAAX (309 cells), GFP-paxillin + mEos2-kindlin-2-QW614/615AA (284 cells), GFP-paxillin + mEos2-kindlin-2-WT-CAAX (289 cells) correspond to pooled data from three independent experiments. Between 13 and 41 fields of view (20x objective) per condition and per experiment were used to quantify cell spreading, except for the cells expressing GFP-paxillin alone (negative control: between 8 and 17 fields per experience). Between 84 and 107 cells per condition and per experiment were included (except for the GFP-paxillin condition: between 31 and 58 cells per experiment). **c** TIRF images of GFP-paxillin showing adhesion sites of Kind[Ko] cells 2 days after re-expression of mEos2-kindlin-2-WT (top) and mEos2-kindlin-2-ΔPH-CAAX (bottom) and 4 h after seeding on fibronectin. For each image, the upper-right panel shows the outlined region at higher magnification. Quantification of number of adhesions per cell (**d**) and total cell area (**e**) of Kind[Ko] cells (4 h after seeding on fibronectin) re-expressing for 2 days mEos2-kindlin-2-WT (light blue), mEos2-kindlin-2-ΔPH (orange), mEos2-kindlin-2-ΔPH-CAAX (green), mEos2-kindlin-2-QW614/615AA (red) or mEos2-kindlin-2-WT-CAAX (dark blue). Each point in the distribution represents the value obtained from a single cell. FAs were drawn manually and cell boundaries were determined by manually setting a threshold on the pixel intensity values using the TIRF GFP-paxillin images as shown in (c). Black bars represent medians and interquartile ranges. GFP-paxillin + mEos2-kindlin-2-WT: 84 cells; GFP-paxillin + mEos2-kindlin-2-ΔPH: 74 cells; GFP-paxillin + mEos2-kindlin-2-ΔPH-CAAX: 62 cells; GFP-paxillin + mEos2-kindlin-2-QW614/615AA: 30 cells; GFP-paxillin + mEos2-kindlin-2-WT-CAAX: 63 cells. The results correspond to pooled data from three independent experiments with at least 8 cells per experiment. Where indicated, statistical significance was obtained using two-tailed, non-parametric Mann–Whitney rank sum test. The exact $P$ values are indicated on the figure except when $P < 0.0001$. Source data are provided as a Source Data file. **f** Schematic model of integrin activation by kindlin and talin in mature FAs. Integrin and kindlin enter independently in FAs by membrane free diffusion whereas talin and paxillin reach FAs by a cytosolic free diffusion. In a FA, integrins display cycles of free-diffusion and immobilizations triggered by binding to kindlin and talin. The probability of freely diffusing integrin, kindlin and talin to meet simultaneously with the correct orientation is extremely low. Our results favor a model where the encounter of freely diffusing integrin and kindlin at the plasma membrane triggers the formation of an immobile integrin-kindlin complex (i). This complex could constrain the integrin β-tail orientation and favor the binding of talin to the proximal NPxY motif, leading to the formation of a transient tripartite integrin-kindlin-talin complex (ii). Then, kindlin could intermittently dissociate (iii) and re-associate (ii) with the longer-lived integrin-talin complex. The role of transient tripartite integrin-kindlin-talin complex, could extend the duration of integrin/talin interactions. Dissociation of both talin and kindlin will end this cycle of integrin immobilization before its next encounter with kindlin.

QW mutant can still partially bind and activate integrins, in accordance with studies showing that kindlin residues other than Q614 and W615 are involved in the interaction with integrins[22,26–28,42,50]. The most dramatic effect was observed for kindlin-2-ΔPH, which displayed almost no full spreading, a large decrease in the fraction of partially spread cells (Fig. 7a, b), and reduced adhesion formation (Fig. 7d, e and Supplementary Fig. 9), supporting the idea that membrane recruitment and/or free diffusion are critical for kindlin-2 functions. Importantly, rescuing membrane recruitment and free diffusion of kindlin-2-ΔPH using the CAAX strategy (Supplementary Fig. 7a–c, e) strongly rescued cell spreading and increased adhesion formation compared to kindlin-2-ΔPH (Fig. 7). Forcing kindlin-2-WT membrane free diffusion by fusion of a CAAX sequence does not enhance further kindlin-mediated spreading and adhesion formation (Fig. 7a–e). Altogether, these results demonstrate that kindlin-2 membrane recruitment and diffusion are key events in the mechanisms of integrin activation.

## Discussion

**sptPALM as a readout for protein functions**. In the present study, tracking of proteins carrying point mutations/deletion combined with functional rescue experiments in genetically engineered cells enabled us to directly link the molecular behavior of a protein with its action in a subcellular compartment. Our results highlighted a correlation between kindlin membrane recruitment/free diffusion and integrin activation during cell spreading and FAs formation. By establishing protein dynamics as a readout of protein functions, this study illustrates how sptPALM is a powerful tool to investigate the mechanisms leading to integrin activation inside living, adherent cells. Moreover, the exquisite sensitivity of single molecule approaches combined with TIRF microscopy revealed that kindlin-2-QW and kindlin-

2-ΔPH displayed a significant immobile fraction inside FAs. These results seem to contradict earlier conventional epi-fluorescence experiments that described a loss of recruitment to FAs for both variants[12,14,20,23,42]. However, in conventional epi-fluorescence, the cytosolic background fluorescence prevents the accurate quantification of protein enrichment in FAs. Using TIRF illumination, we revealed that inhibiting kindlin-2 interactions with integrins (kindlin-2-QW) or membrane recruitment and diffusion (kindlin-2-ΔPH) does not totally suppress kindlin-2 recruitment to FAs. However, these molecular events associated synergistically to lead to high and specific local enrichment of kindlin inside FAs. This suggests that kindlin-2 membrane recruitment and diffusion are pivotal prerequisites to drive accumulation of kindlin-2 in FAs via its interactions with integrins[20], ILK[20], or paxillin[17].

**Kindlin-2 and integrin activation in early versus mature adhesions**. Recent studies have highlighted the role of kindlin during early cell spreading and initiation of integrin adhesions[16–18]. Quantification of protein recruitment to NAs, using fluorescence fluctuation methods, demonstrated that kindlin is binding to integrins before talin[16]. The priming role of kindlin is also supported by experiments performed in kindlin- or talin-KO fibroblasts, showing that only kindlin and not talin can induce early spreading and formation of NAs upon $Mn^{2+}$-mediated integrin activation[18]. However, our results indicate that kindlin is not only crucial for integrin activation in NAs but also in mature FAs. Indeed, kindlin controls integrin immobilization in FAs, suggesting an important role for FAs maintenance. Moreover, the level of cell adhesion induced by kindlin correlates with its ability to diffuse along the plasma membrane and to accumulate in mature FAs. We have previously shown that the landscape of immobile integrins in FAs is remodeled in less than

a hundred seconds through cycles of integrin free diffusion and immobilization[6,36] (Fig. 7f). Kindlin could favor integrin activation/immobilization by different mechanisms. First, kindlin could initiate integrin immobilization, by itself activating integrin or by favoring the establishment of integrin-talin interactions (Fig. 7f). These mechanisms thought to control NAs life-cycles in the lamellipodium[16–18] could also be at play to control the life-cycles of integrin immobilizations in mature FAs. Second, kindlin could also act by sustaining integrin activation. Importantly, our current results show that indeed kindlin controls the duration of integrin immobilization in FAs. To do so, kindlin could stabilize integrin-talin interactions, for instance by competing with integrin inhibitors such as ICAP or filamin, known to inhibit talin binding to integrin[51,52]. Indeed, ICAP-1 and filamin binding domains on integrin β-tails are overlapping with kindlin binding domain[10,11]. Also, as we found that paxillin is immobile in FAs, kindlin could also maintain integrin immobilization by coupling integrins to paxillin during transient integrin-talin unbinding events.

**Integrin activation by kindlin-2 in FAs is integrin selective.** Different integrin classes are present in the same mature FAs and use distinct mechano-transduction and signaling pathways that cooperate to control FAs structure and functions, such as migration, rigidity sensing and signaling. In a previous study, we demonstrated a nanoscale horizontal partitioning and specific dynamics for integrin α5β1 and αvβ3 within FAs; β3-class integrins are stationary and enriched within FAs, whereas β1-class integrins are less enriched and display rearward movements[6]. Furthermore, the use of fibronectin micropatterned substrates emphasized the distinct localization of αv- or β1-class integrins in FAs[53]. These results indicated that specific classes of α/β integrins (α5β1 and αvβ3) act as distinct 'nanoscale adhesion units' within an individual FA with specific dynamics, organization and force transmission of F-actin motion to the ECM. Altogether these findings suggest that nanoscale-partitioning of integrins in FAs reflects the segregation of specific signaling properties and mechanical tasks that cooperate to determine the functions of FAs. Inhomogeneous localization of αv-or β1-class integrins implies that their associated signaling proteins are also probably segregated. In line with this hypothesis, a proteomic study showed that β1-class integrins link kindlin-2 and the ILK/Pinch/Parvin complex to myosin II activation[53], indicating that kindlin-2 is preferentially coupled to the functions of β1-integrin rather than β3-integrin. This is consistent with our sptPALM results, which show that the fraction and dwell-time of immobilized β1-integrins in mature FAs are more sensitive to kindlin-2 binding than immobilized β3-integrins. Indeed, we observed that an integrin mutation decreasing interaction with kindlin-2 has more effects on β1-integrins than β3-integrins either inside and outside FAs. Furthermore, $Mn^{2+}$ treatment triggering cell spreading (i.e. early integrin activation) depends on the presence of kindlin[18]. However, our results revealed that $Mn^{2+}$-induced integrin immobilizations is not impaired for β3-759A (kindlin mutant) but is impaired for β1-795A (kindlin mutant), suggesting again that β1-integrin activation critically relies on kindlin-2. In addition, αvβ3- and α5β1-integrin abilities to sense and transmit forces are distinct. Increased application of extracellular forces causes strengthening of linkages between αvβ3-integrin and F-actin cytoskeleton in a talin-dependent fashion and thus rigidity sensing, while the strength of the α5β1-integrin/F-actin connection seems unaffected by force generation[54–57]. On the extracellular side, α5β1-integrin binding to FN is stabilized in a force-dependent conformational transition[58,59]. This could explain why α5β1-integrin/ECM bonds generate and resist to higher forces compared to αvβ3-integrin/ECM bonds[53,57]. An interesting model could be that the activation of β1-integrin is mainly fostered by biochemical

reactions such as kindlin-2 binding while the activation of αvβ3-integrin could be mainly promoted by force generation via talin.

**Is it better to jump or to crawl?** What would be the functional advantage of two-dimensional (2D) membrane diffusion of kindlin over three-dimensional (3D) cytosolic diffusion? Constraining the movements of proteins from a 3D to a planar 2D system could affect both the binding affinities between two molecules[60] and the frequency of encounter[61,62]. Our results clearly showed that kindlin membrane diffusion promotes integrin activation. First, the membrane anchoring topology of kindlin may constrain its orientation, raising the frequency of productive interactions with integrins[60,63]. Second, membrane diffusion of kindlin funnels kindlin random molecular motion to the location where integrins reside, which might increase the probability of encounter. However, membrane 2D diffusion (0.1–1 μm²/s) is two orders of magnitude slower than cytosolic 3D diffusion (10–100 μm²/s). An exhaustive 2D search would thus imply an increased searching time. However, this searching time can be reduced by increased concentrations of integrins and kindlins, such as found inside FAs. The searching time can also be reduced by intermittent free diffusion in the cytosol[62], which could be at play outside FAs where the concentration of kindlins and integrins are low. Such intermittent 2D scans and 3D motions were observed for PH and C2 domains using SPT[64,65]. Nevertheless, our results suggest that a large fraction of full-length kindlin is associated with the plasma membrane. This could be explained by the multiple membrane binding domains of kindlin-2[12,23,29,41,43]. Using functional assays, we showed that forcing kindlin membrane free diffusion by fusion of a CAAX sequence does not enhance nor diminish kindlin action, suggesting that the functional advantage of kindlin membrane diffusion is already at its maximum for wild-type kindlin, and that intermittent 3D motions are not required for efficient integrin activation by kindlin.

**Distinct ways for searching the same target at the membrane: lessons from kindlin and talin.** The direct observation of kindlin movements in living adherent cells demonstrates that talin and kindlin are recruited to FAs by different parallel pathways. Talin is recruited directly from the cytosol without prior membrane association[6], whereas kindlin recruitment to FAs relies on lateral membrane diffusion followed by trapping in FAs. The distinct pathways used by talin and kindlin to reach integrins could constitute the molecular basis for their complementarity during integrin activation. Talin in FAs adopts a polarized and extended conformation with the head located down in the integrin layer and the tail located in the upper actin regulatory layer by binding to actin and vinculin[4,5,8]. Talin tail tethering to actin filaments could maintain talin in FAs during release of its autoinhibition, and could enable talin head to be at the proximity of membrane constituents such as PIP2 during integrin activation[66–69]. Results obtained with 3D super-resolution microscopy, using DONALD[48], indicate that kindlin and paxillin are located at the same height in FAs, where talin head also resides[4]. Thus, kindlin is essentially found in the integrin layer, reinforcing the idea that membrane recruitment and diffusion directly support kindlin functions. In the same line, kindlin binding to paxillin, which is located in the integrin layer, probably helps maintaining high pool of kindlins at the proximity of integrin tails to increase integrin activation[18,23,26]. As the probability for three proteins to meet simultaneously with the correct orientation is extremely low, we favor a model with the sequential formation of an immobile integrin-kindlin complex followed by the formation of a transient

tripartite integrin-kindlin-talin complex where kindlin could be replaced by talin head and reciprocally (Fig. 7f). Indeed, as kindlin-2 and talin-1 immobilizations in FAs correspond in part to interactions with immobilized integrins, the shorter duration of kindlin-2 immobilizations suggests that kindlin-integrin interactions are shorter than talin-integrin interactions. This also suggests a low occurrence of stable tripartite integrin-kindlin-talin complexes, otherwise the immobilization durations would have been similar for kindlin and talin. These results rather support the existence of transient tripartite integrin-kindlin-talin complexes, either enabling the initiation of integrin-talin interactions, or extending the duration of integrin/talin interactions. The immobile integrin-kindlin complex could constrain the integrin β-tail orientation and favor the binding of talin to the proximal NPxY motif.

Proteins co-evolved to achieve increasingly difficult tasks such as the formation of functional macromolecular complexes. This study illustrates how protein domains convert random molecular motions into sequences of molecular events leading to the formation of elaborate macromolecular structures, which are essential for cellular functions such as adhesion and motility. Among all the possible protein-protein interactions that can occur in a protein interaction network, such as the integrin adhesome[2], specific single molecule behavior of proteins determines both the location and timing of possible interactions, and can thus assist protein functions by funneling proteins on their targets. The simultaneous orchestration of these molecular ballets is yet to be understood.

## Methods

**DNA constructs and antibodies.** mEos2-kindlin-2-WT (human, vector: pcDNA3, promoter: CMV), kindlin-2-ΔPH (human, vector: pcDNA3), mEos2-kindlin-1 (human, vector: pcDNA3, promoter: CMV) and GFP-kindlin-2-WT (human, vector: pcDNA3, promoter: CMV) plasmids were gifts from B. Wehrle-Haller (U. Genève, Suisse). The mEos2-kindlin-2-ΔPH construct (human, vector: pcDNA3, promoter: CMV) was obtained by removing the kindlin-2-WT sequence in the mEos2-Kindlin-2-WT and inserting the kindlin-2-ΔPH sequence by endonuclease cleavage at EcoNI/NotI sites. mEos2-kindlin-2-WT-CAAX construct (human, vector: pcDNA3, promoter: CMV) was obtained by inserting a 6 aminoacid (GSSGSS) followed by a CAAX box sequence in the mEos2-Kindlin-2-WT at BsrGI/BamHI endonuclease sites. mEos2-kindlin-2-ΔPH-CAAX (human, vector: pcDNA3, promoter: CMV) was obtained by inserting a 6 amino-acid (GSSGSS) followed by a CAAX box sequence in the mEos2-kindlin-2-ΔPH at EcoRI endonuclease sites. mEos2-PH-domain of kindlin-2 construct (human, vector: pcDNA3, promoter: CMV) was obtained by replacing the kindlin-2-ΔPH sequence in the mEos2-kindlin-2-ΔPH construct by a sequence containing the PH domain and parts of the F2 subdomains of kindlin-2 (K341 to D504). mEos2-kindlin-2-QW, mEos2-kindlin-2-L357A, mEos2-kindlin-2-K390A mutants (human, vector: pcDNA3, promoter: CMV) were generated by site-directed mutagenesis using the QuickChange II XL Site-Directed Mutagenesis kit (Stratagene) on the mEos2-kindlin-2-WT plasmid described above. Talin1-tdEos was a gift from P. Kanchanawong (MBI, NUS, Singapore). For the mEos2-tagged human beta1-integrin construct (vector: pFB), the mEos2 sequence (Addgene) was PCR-amplified and used to replace the GFP of pFB-Neo-human beta1-integrin-GFP (gift from M. Humphries-University of Manchester, UK) to generate an in-frame C-terminal fusion. Human beta3-integrin-GFP plasmid (vector: pEGFP-N1), was provided by N. Kieffer (CNRS/CRP, Luxembourg). Human 6His-beta3-integrin-GFP was obtained from the previous one by domain swapping to introduce the 6His tag and glycine linker in the N-terminus. For the human 6His-beta3-integrin-mEos2 (vector: pEGFP-N1), a PCR of mEos2 was done on pRSETa-mEos2 (Addgene) to replace the GFP from human 6His beta3-integrin-GFP at the AgeI/BsrGI sites. Beta3-integrin-Y747A, beta3-integrin-Y759A and beta1-integrin-Y783A were generated by site-directed mutagenesis using the QuickChange II XL Site-Directed Mutagenesis kit (Stratagene) on the 6His-beta3-integrin-mEos2 described above and the mEos2-tagged human beta1-integrin[6] respectively. Beta1-integrin-Y795A-mEos2 construct (human, vector: pFB) was kindly provided by C. Albiges-Rizo (IAB, U. Grenoble Alpes). mEos2-paxillin construct (chicken, promotor: CMV, vector: pmEos2) was obtained from Addgene (Plasmid #57409). The cytosolic mEos2 construct was obtained by inserting the mEos2 sequence of the prSETa mEos2 plasmid (Addgene) in a pcDNA vector at BamHI / EcoRI endonuclease sites. The fidelity of all constructs was verified by sequencing.

For DONALD, we used GFP-tagged proteins that we imaged by dSTORM using a home-made anti-GFP nanobody labeled with AlexaFluor-647 used at 85 nM[49].

For western blots, we used a monoclonal anti-kindlin-2 antibody (Merck, Cat# MAB2617, Clone 3A3, Lot# 3114519) and a monoclonal anti-β-actin antibody (Sigma-Aldrich, Cat# A5316, Clone AC-74).

**Cell culture and sample preparation.** Mouse Embryonic Fibroblasts (MEFs) and kindlin-1, kindlin-2 double knock-out cells (Kind$^{Ko}$) were cultured in DMEM (Gibco) with 10% FBS (Gibco), GlutaMAX supplement, 100 U.ml$^{-1}$ penicillin-streptomycin, 1 mM sodium pyruvate, 15 mM HEPES. Transient transfections of plasmids were performed 1–3 days before experiments using the Nucleofector™ transfection kit for MEF-1 and Nucleofactor™ IIb device (Amaxa™, Lonza). The cells were detached with 0.05% trypsin, 0.02% EDTA solution (Gibco). The trypsin was inactivated using soybean trypsin inhibitor (1 mg/ml in DMEM, Sigma), and the cells were washed and suspended in serum-free Ringer medium (150 mM NaCl, 5 mM KCl, 2 mM CaCl$_2$, 2 mM MgCl$_2$, 10 mM HEPES, pH=7.4) supplemented with 11 mM glucose. Cells were then seeded on human fibronectin-coated surface (fibronectin: 10 µg/ml, Roche).

Cells were co-transfected with mEos2-fused proteins and GFP-paxillin as a FA reporter. To clearly discriminate NAs, initiated 0.5 µm back from the lamellipodium tip[70,71] from mature FAs, resulting from the subsequent maturation of a fraction of NAs outside of the lamellipodium, we used MEFs spread on fibronectin for more than 3 h possessing well-defined mature FAs. Region of interests where chosen outside of active lamellipodia where NAs are located. For results obtained in early NAs (Supplementary Fig. 3), we performed experiments in active lamellipodia during cell spreading, between 15 and 60 min after cell loading[70,71].

The Kind$^{Ko}$ cell line was kindly provided by Reinhard Fässler (Max Planck Institute of Biochemistry, Martinsried) and are described elsewhere[18]. Absence of mycoplasma contamination was assessed using the MycoAlert detection kit (Lonza). For sptPALM, DONALD and membrane fraction assays, 50,000 cells were seeded on fibronectin-coated #1.5H glass coverslips (Marinfield). For cell spreading assays, 5000 cells were seeded on fibronectin-coated standard plastic 12 well culture plates. For focal adhesion measurements, 3000 cells were seeded on fibronectin-coated glass bottom 96 well plates (SCHOTT).

**Western blot.** To assess kindlin-2 expression levels, transfected MEF cells were lysed after 2 days in lysis buffer: Hepes (800 mM) 50 mM + EDTA (0.5 M) 0.5 mM + EGTA (0.5 M) 4 mM + NaCl (3 M) 150 mM + Triton 1%, containing 1X protease inhibitor cocktail (Calbiochem/Millipore). Proteins concentrations were measured using Bradford assay and 10 µg of total protein were mixed with SB2X in equal volume. Proteins were loaded on pre-cast gel (Biorad gradient gel 4–20%, 15 wells - ref: 456,1096) after heating at 95 ℃ for 5 min. Protein size marker (Thermo Scientific- Pageruler Prestained plus – ref: 26619) was loaded without heating. The electrophoresis was run at 200 V, 3 A for 40 min in the running buffer and proteins separated on the gel were transferred to PVDF membrane at 100 V, 3 A for 1 h. The membrane was blocked for 30 min at room temperature with a BSA solution, 5% in TBS and after washes incubated with a primary antibody on a rocking platform at room temperature for 2 h. The primary antibody was diluted with 5% BSA in 0.1%Tween TBS: 20 mM Tris pH 7,5 and 137 mM NaCl, for Kindlin2 (MAB2617- Merck) (1/1000) and β-actin (A5316- Sigma) (1/1000). After 5 washes in TBS, the membrane was incubated with a secondary antibody diluted (1/10,000) with 5% BSA in 0.1% Tween TBS for 45 min at room temperature. After washes, the membrane was incubated for 3 min under agitation with 2 reagents vol/vol (SuperSignal™ West Femto Maximum Sensitivity Substrate, pierce, ref: 34095). The membrane was revealed on a Chemidoc setup with Image Lab Software. Western Blot quantifications were obtained using ImageJ.

**Integrin activation with manganese.** When mentioned, integrin activation was induced by replacing the cell media (Ringer + glucose) with a Ringer+glucose solution with MnCl$_2$ at 5 mM, at least 5 min before acquisition.

**Microscopy acquisitions.** All acquisitions (except for DONALD modality) were steered by MetaMorph software (Molecular Devices) with an inverted motorized microscope (Nikon Ti) equipped with a temperature control system (The Cube, The Box, Life Imaging Services), a Nikon CFI Apo TIRF 100x oil, NA 1.49 objective (except cell spreading assays: Nikon CFI Plan Apo Lambda DM 20x objective), and a perfect focus system, allowing long acquisition in TIRF illumination mode. When cells were seeded on coverslips, the coverslip was mounted in a Ludin chamber (Life Imaging Services) before acquisition.

**Single particle tracking photo-activated localization microscopy (sptPALM)**
*Optical setup and image acquisition.* Imaging was performed at least 4 h after seeding the cells on fibronectin-coated coverslips. For photoactivation localization microscopy, cells expressing mEos2 tagged constructs were photoactivated using a 405 nm laser (Omicron) and the resulting photoconverted single molecule fluorescence was excited with a 561 nm laser (Cobolt Jive™). Both lasers illuminated the sample simultaneously. Their respective power was adjusted to keep the number of the stochastically activated molecules constant and well separated during the

acquisition. Fluorescence was collected by the combination of a dichroic and emission filters (D101-R561 and F39-617 respectively, Chroma) and a sensitive EMCCD (electron-multiplying charge-coupled device, Evolve, Photometric). The acquisition was performed in streaming mode at 50 Hz. GFP-paxillin was imaged using a conventional GFP filter cube (ET470/40, T495LPXR, ET525/50, Chroma). Using this filter cube does not allow spectral separation of the unconverted pool of mEos2 from the GFP fluorescent signal. However, with all of the constructs used, whether the mEos2 signal was highly or poorly enriched in FAs, we were still able to detect FAs with GFP-paxillin.

**Single molecule segmentation and tracking**. A typical sptPALM experiment leads to a set of at least 4000 images per cell, analyzed in order to extract molecule localization and dynamics. Single molecule fluorescent spots were localized and tracked over time using a combination of wavelet segmentation and simulated annealing algorithms[72–74]. Under the experimental conditions described above, the resolution of the system was quantified to 59 nm (Full Width at Half Maximum, FWHM). This spatial resolution depends on the image signal-to-noise ratio and the segmentation algorithm[75] and was determined using fixed mEos2 samples. We analyzed 130 2D distributions of single molecule positions belonging to long trajectories (>50 frames) by bi-dimensional Gaussian fitting, the resolution being determined as $2.3 s_{xy}$, where $s_{xy}$ is the pointing accuracy.

For the trajectory analysis, FAs ROIs were identified manually from GFP-paxillin images. The corresponding binary mask was used to sort single-molecule data analyses to specific regions. We analyzed trajectories lasting at least 260 ms (≥13 points) with a custom Matlab routine analyzing the mean squared displacement (MSD), which describes the diffusion properties of a molecule, computed as (1):

$$MSD(t = n \cdot \Delta t) = \frac{\sum_{i=1}^{N-n} (x_{i+n} - x_i)^2 + (y_{i+n} - y_i)^2}{N - n} \quad (1)$$

where $x_i$ and $y_i$ are the coordinates of the label position at time $i \times \Delta t$. We defined the measured diffusion coefficient $D$ as the slope of the affine regression line fitted to the $n$=1 to 4 values of the MSD ($n \times \Delta t$). The MSD was computed then fitted on a duration equal to 80% (minimum of 10 points, 200 ms) of the whole stretch by (2):

$$MSD(t) = \frac{4r_{conf}^2}{3}\left(1 - e^{-t/\tau}\right) \quad (2)$$

where $r_{conf}$ is the measured confinement radius and $\tau$ the time constant $\tau = (r_{conf}^2/3D_{conf})$. To reduce the inaccuracy of the MSD fit due to down sampling for larger time intervals, we used a weighted fit. Trajectories were sorted in 3 groups: immobile, confined diffusion and free diffusion. Immobile trajectories were defined as trajectories with $D < 0.011$ µm².s⁻¹, corresponding to molecules which explored an area inferior to the one defined by the image spatial resolution ~(0.05 µm)² during the time used to fit the initial slope of the MSD[6] (4 points, 80 ms): $D_{threshold} = (0.059$ µm)²/(4 × 4 × 0.02 s)~0.011 µm².s⁻¹. To separate trajectories displaying free diffusion from confined diffusion, we used the calculated time constant $\tau$ for each trajectory. Confined and free diffusion events were defined as trajectories with a time constant respectively inferior and superior to half the time interval used to compute the MSD (100 ms).

With the acquisition frequency and TIRF illumination used in our experiments, it is impossible to reconnect trajectories lasting at least 260 ms (≥13 points) for a protein freely diffusing in 3D within the cytosol. Therefore, the very low occurrence of free-diffusing trajectories inside or outside FAs for kindlin-2-ΔPH and paxillin indicates that these proteins are mostly moving in the cytosol.

*Kymographs generation and analysis*. Using sptPALM acquisition (50 Hz, >12,000 frames—240 s) in which fluorescent single molecules were localized as described above, we generated super-resolution PALM time-lapse sequences in which each frame corresponds to a merge of 25 super-resolution frames (resulting image duration: 0.5 s). In these time-lapses, high-density zones of detections corresponded to immobile mEos2. Kymographs were generated from lines tangential to focal adhesions and analyzed using the ImageJ plugin KymoToolBox (F. Corde-lières). The duration of mEos2 immobilization was then determined by manually measuring the lengths of high-density zones (Figs. 1k, 2h). Immobilized mEos2 are not continuously emitting light but instead repeatedly switch between non-emmisive and emissive states, a photophysical phenomenon called "blinking"[76]. In this quantification, it is therefore possible that for some events, a persisting spot of immobile mEos2 detections was erroneously considered as originating from a single molecule, which would induce an overestimation of immobilization durations. On the other hand, immobilization duration of single mEos2 molecules may be underestimated due to photobleaching. However, as comparable illumination settings (fixed 561 nm laser power, varying 405 nm laser power) were used between the different experimental conditions, the strong variations in immobilization durations presented here cannot be due to blinking or photobleaching.

**Direct optical nanoscopy with axially localized detection (DONALD)**. This method combines super-localization microscopy techniques with detection of supercritical-angle fluorescence emission. This analysis provides an absolute measure of the vertical position of the fluorescent emitter regarding the coverslip. The axial resolution of DONALD enables to localize with 35 nm resolution proteins in the different functional layers.

**Optical setup**. We used an Olympus IX83 inverted microscope with an autofocus system. The excitation path was composed of three laser lines: 637 nm, 532 nm, and 405 nm (Errol lasers) and a TIRF module (Errol lasers) used in combination with a matched 390/482/532/640 multiband filter (LF405/488/532/635-A-000, Semrock). The fluorescence was collected through an Olympus x100 1.49 NA oil immersion objective lens. The detection path was composed of a SAFe module (Abbelight) and a Flash 4 v3 (Hamamatsu). The pixel size in the resulting image was 100 nm.

**Image acquisition**. The diffraction limited epifluorescence images were acquired at low illumination irradiance (0.15 kW.cm⁻²), while dSTORM images were obtained using a high illumination irradiance (4 kW.cm⁻²) until a sufficient molecule density was obtained (around 1 molecule per µm²) and the acquisition could be started. The exposure time was set at 50 ms, optimal timing with the buffer to capture all emitted photons in a single frame. All the acquisitions were performed using the Nemo software (Abbelight). To achieve single molecule regime in dSTORM acquisitions, a dedicated buffer (Smart kit, Abbelight) was used.

**Image processing**. Acquired data were processed using the Nemo software (Abbelight). After removing the background signal, molecules were detected and the numbers of epifluorescence and under-critical angle fluorescence (UAF) photons were measured to extract the corresponding axial positions as described elsewhere[48]. Lateral drifts were corrected from the localized data thanks to a cross-correlation based algorithm. DONALD benefits from the unique property of being free of any axial drift, as the supercritical emission allows one to extract the absolute axial position of the fluorophore regarding the coverslip/sample interface.

**Image display**. Molecules detected 150 nm above the surface were discarded to improve the contrast in the focal adhesions/plasma membrane layer. On the super-resolution reconstructed image, each pixel value corresponds to the average axial localization of single molecules detected in this pixel (size: 15 nm). For ease of observation, the obtained images were smoothed using a xy mean filter with a 5 × 5 kernel.

**Z distributions**. For the curve of occurrence (Fig. 5f, g), every detections were included. For the box plot (Fig. 5h), molecules detected above 200 nm were discarded to improve the relevance of the displayed median and average z positions.

**Assessment of protein membrane fraction**. Imaging was performed at least 3 h after seeding the cells on fibronectin-coated coverslips. For a given protein, the proportion of membrane associated-protein was estimated in several cells by comparing the membrane-level fluorescence signal (TIRF) with the total fluorescence signal of the cell (epifluorescence) as done previously[44,45]. For each cell, cell boundaries were determined using the TIRF image by manually setting an intensity threshold, or by manual region drawing. The amount of membrane associated protein ($n_{membrane}$) was then estimated by the mean fluorescence measured in this region (Fluo$_{TIRF}$), divided by the average background fluorescence (BG$_{TIRF}$) measured in 3 different locations on the image:

$$n_{membrane} = \frac{Fluo_{TIRF}}{BG_{TIRF}}$$

The total amount of protein in the cell was assessed by doing these measurements in the epifluorescence image. The total amount of protein ($n_{total}$) was thus estimated by the mean fluorescence measured in the cell region (Fluo$_{epi}$), divided by the average background fluorescence (BG$_{epi}$), using each time the same "cell" and "background" regions as those used in the related TIRF image:

$$n_{total} = \frac{Fluo_{epi}}{BG_{epi}}$$

The membrane fraction of the protein in the cell ($F_{membrane}$) was estimated by the ratio of membrane protein ($n_{membrane}$) to total protein ($n_{total}$):

$$F_{membranaire} = \frac{n_{membrane}}{n_{total}}$$

TIRF and epifluorescence images were obtained using the photoconverted mEos2 signal with the same optical set-up used for sptPALM experiments. To perform ensemble mEos2 measurements, each cell was illuminated by a 405 nm laser in epifluorescence mode for 4 s just before the acquisition to ensure massive and cell-wide mEos2 photoactivation.

**Focal adhesion enrichment**. Imaging was performed at least 3 h after seeding the cells on fibronectin-coated coverslips. Segmentation of focal adhesion was performed on the GFP-paxillin TIRF images using a custom wavelet-based image

segmentation and/or the Focal Adhesion Analysis Server detection system (http://faas.bme.unc.edu)[77] and completed by manual region drawing when necessary. Recruitment to FA of Kindlin-2 (WT or mutated variants) coupled to mEos2, referred to as "focal adhesion enrichment", was defined as:

$$FA\ enrichment = \frac{average\ fluorescence\ intensity\ inside\ FA - average\ background\ intensity}{average\ fluorescence\ intensity\ outside\ FA - average\ background\ intensity}$$

where "average fluorescence intensity" is the signal of the photoconverted mEos2 using the same materials as used during sptPALM experiments, and "average background intensity" refers to the signal measured outside the cell. For each cell, cell boundaries were defined either by manually setting a threshold on the pixel intensity values, or by manual region drawing. To ensure a massive, homogeneous photoconversion of the mEos2, each cell was illuminated by a 405 nm laser in epifluorescence mode for 4 s just before the acquisition.

**Cell spreading assays**. Phase contrast images were recorded at least 4 h after seeding the cells on fibronectin-coated 12 well culture plates. Between 13 and 41 fields of view (20x objective) per condition and per experiment were used to quantify cell spreading, except for the cells expressing GFP-paxillin alone (negative control: between 8 and 17 fields per experience). Non-isolated cells (i.e., in contact with other cells) and polynucleated cells were excluded. Transfection efficiency was checked with the GFP-paxillin signal. Cells showing a GFP-paxillin average signal below 1.5-fold the average background fluorescence signal were excluded. Between 84 and 107 cells per condition and per experiment were included (except for the GFP-paxillin condition: between 31 and 58 cells per experiment). Cells were partitioned into three categories: (i) non-spread when round; (ii) partially spread when at least one large protrusion was contacting the substrate, or when the cell body was partially deformed compared to spheroid non-adhering cells; and (iii) fully spread when the cell area was highly superior to the mean cell area, or when the cell body was not spheroid and the cell was polygonal rather than filiform.

**Focal adhesion measurements**. Imaging was performed at least 4 h after seeding the cells on fibronectin-coated glass bottom 96 well plates (SCHOTT). Single cells are then chosen randomly and the fluorescence of GFP-paxillin is recorded at high magnification (100x) in TIRF settings for adhesion site measurements. The expression of the mEos2-kindlin-2 or mutated variants was checked for each cell. Adhesion sites were drawn manually, and quantification of adhesions sites were computed using MetaMorph software (Molecular Devices). Non-isolated cells (i.e., in contact with other cells) and polynucleated cells were excluded. Between 23 and 42 cells per experiment were used for the kindlin-2-ΔPH or kindlin-2-ΔPH-CAAX conditions (kindlin-2-WT: between 17 and 26 cells per experiment). For each cell used for focal adhesion measurements, the cell area was measured using the TIRF image of GFP-paxillin.

**Statistical analysis**. The indicated $P$ values were obtained with the two-tailed, non-parametric Mann–Whitney rank sum test using the software GraphPad Prism 8. For z distributions, the Mann–Whitney rank sum test was performed using Matlab 2007a due to very high sample size. The exact $P$ values are indicated on all the main and supplementary figures except when $P < 0.0001$.

**Reporting summary**. Further information on research design is available in the Nature Research Reporting Summary linked to this article.

## Data availability

Data supporting the findings of this manuscript are available from the corresponding authors upon reasonable request. A reporting summary for this Article is available as a Supplementary Information file. Source data are provided with this paper.

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

## Acknowledgements

We thank B. Tessier, R. Sterling, J. Carrere for technical assistance and the IINS Cell culture facility, especially E. Verdier, for cell culture; A. Mehidi, M. Lagardère, C. Saphy for helpful discussions; C. Poujol, S. Marais (Bordeaux Imaging Center, BIC) for technical help; F. Cordelières (BIC) for support in kymograph analysis (KymoTool Box). We thank N. Bourg and C. Schietroma (Abbelight) for providing DONALD acquisition system with dSTORM dedicated buffer (Smart kit) and help for the DONALD data analysis. We thank M. Sainlos (IINS, Bordeaux, France) for providing anti-GFP nanobodies labeled with AlexaFluor-647 used for DONALD experiments; C. Albiges-Rizo (IAB, U. Grenoble Alpes) and P. Kanchanawong (MBI, NUS, Singapore) for providing the beta1-integrin-Y795A-mEos2 and the talin1-tdEos plasmid construct, respectively. We acknowledge financial support from the French Ministry of Research and CNRS, GDR MIV-ImaBio Bourse AMI (to O.R.), Fondation pour la Recherche Médicale (to T.O./O.R. FDM20140630221), Fondation pour la Recherche Médicale (to G.G.), LABEX Brain (to T.O.), ANR grant Integractome (to G.G.).

## Author contributions

G.G. and O.R. conceived and coordinated the project. G.G., O.R., and T.O. conceptualized the experiments. T.O. and O.R. performed and analyzed sptPALM experiments. J-B.S. provided analytical tools for sptPALM experiments. O.R., T.O., C.C., and S.L-F. performed and analyzed DONALD experiments. T.O. and A.J. performed and analyzed membrane and FAs recruitment experiments. R.F. and R.B. generated Kind[Ko] cells and T.O. and A.J. performed the associated rescue experiments. Z.K., B. W-H., and B.K designed and generated protein constructs. Z.K. performed western blot experiments. G.G., O.R., and T.O. wrote the manuscript with input from all authors.

## Competing interests

S.L-F. declare the existence of a financial competing interest as S.L-F. is shareholder in the Abbelight company developing DONALD, the 3D super-resolution modality. All other authors declare no competing interest.
