## [Peer Review File · Nature Communications]

Reviewer #1 (Remarks to the Author):

In the present manuscript, T. Orré and colleagues use single particle tracking and 3D nanoscopy to address the localization and dynamics of kindlin 2 in focal adhesions. In particular, they reveal that kindlin is important for integrin immobilization at focal adhesions and that, contrarily to talin, it freely diffuses at the plasma membrane. Furthermore, they demonstrate the importance of its PH domain for its localization. This very complete, well-documented and clear manuscript constitutes an important source of information to understand the role of kindlin 2 in cell adhesion. However, some information could be added and clarified to strengthen the impact of the manuscript.

Main comments:

-The authors chose to explore the function of kindlin in mature focal adhesions only, and not in nascent adhesions. However, it is not specified how mature adhesions, and not nascent adhesions, were selected for the analysis. Furthermore, as kindlin was shown to be important for the formation of nascent adhesions (Bachir et al. 2014; Theodosiou et al. 2016; Böttcher et al. 2017), it would also be interesting to address the dynamics of kindlin in nascent adhesions.

-Fig.4d shows that the PH domain alone displays the same fraction of immobilization than K2-WT. This seems inconsistent with Fig3 that would argue for a dependency on interaction with integrin for kindlin immobilization. This should be commented.

-In Fig.5, the authors detail the nanoscale 3D localization of kindlin in focal adhesions, taking as references paxillin. First, there is a small discrepancy in the axial localization of paxillin compared with the results published by Kanchanawong et al. (Nature 2010). Second, according to Kanchanawong et al., Kindlin being located at 48.5 nm above the coverslip, it should be considered in the “force transduction layer”, rather than in the integrin layer. Since there seems to be some differences with the work of Kanchanawong et al., I suggest to determine the axial localization of other markers of the different layers of focal adhesions (e.g. CAAX, integrins, vinculin and actinin) to be able to conclude about the layer in which kindlin is localized.

-It is not clear whether the 3D nanoscopy analysis was performed only in focal adhesions or in the whole cell edges. As for the other figures, I also suggest to perform these measurements both inside and outside adhesions.

Minor comments:

-The introduction of this manuscript lacks an introduction on the current knowledge on the function of kindlins at focal adhesions.

-It should be indicated that beta3-Y747A is not specific of talin binding as it also decreases the interaction with other proteins, including FlnA (Petrich et al. 2007).

-I propose to discuss the possible dimerization of kindlin, and in particular how it could affect kindlin diffusion.

-Result/ Disruption of the integrin-kindlin interaction increases kindlin diffusion inside FAs: “These results support the idea that kindlin controls integrin immobilization inside mature FAs.” Shouldn't it

be written: "These results support the idea that integrin controls kindlin immobilization inside mature FAs." ?

-O. Rossier et al. have previously shown that talin is partly moving rearward in focal adhesions. It would be interesting to address whether it is also the case for kindlin, in order to evaluate to what extent it is coupled to the actin retrograde flow.

-Fig.5h: I suggest to represent data similarly to Fig.4a-c from Kanchanawong et al., and to display orthogonal sections of FAs.

-The authors propose in the discussion that kindlin could stabilize integrin-talin interactions. This is an important question that could be addressed by determining the effect of kindlin KO on talin diffusion.

-It would help the reader to add a cartoon that summarizes the findings of this study.

Reviewer #2 (Remarks to the Author):

In this study, live-cell single-particle tracking (SPT) and super-resolution microscopy techniques were applied to study the motion characteristics of kindlin in relation to its localization to focal adhesions (FA). Based on site-specific mutations, integrin- β 1 and β 3 immobilization in FAs is shown to be dependent on the binding of their cytoplasmic domains to kindlin. Kindlin-1 and -2 were shown to undergo primarily lateral diffusion in the membrane plane, in contrast to Talin which enter FA largely via 3D diffusion. Kindlin-2 immobilization in FA depends on its interaction with activated integrin, but not paxillin or ILK. Kindlin-2 membrane diffusion is mediated by PH domain and interaction with phosphoinositides, and is important for nanoscale targeting to the membrane-proximal compartment. Kindlin association with the membrane is shown to be important for FA localization and enrichment, and promote cell spreading.

Overall this is a rigorous study using state-of-the-art techniques and KO cell lines established earlier by the authors. The approach gives mechanistic insight that is highly informative on the function of kindlin in cell adhesions. Data is well-presented and the experiments are well-controlled. I am supportive of its eventual acceptance to Nature Communications. However there is one important point that should be addressed.

1. The Calderwood lab showed in 2014 (Huet-Calderwood et al. J Cell Sci) that ILK and Kindlin-2 interactions appeared to be important for kindlin targeting to FAs. However, this seems to contrast with the results in Fig. S3 in this study where the ILK-binding mutant of kindlin, L357A shows essentially the same behaviors as WT. In the Calderwood study, L357A kindlin-2 seems to have FA-localization defect. However, in inset of Fig. S3a, the FA localization of L357A kindlin-2 seems to be quite substantial.

While the authors may have mentioned these localization differences in passing, given the extensive and rigorous works on ILK, Kindlin-2, and FAs reported earlier by the Calderwood lab, I feel that any differences on ILK/Kindlin-2 interactions observed here should be looked into or discussed at some depth in context of these earlier findings.

Minor points:

- It would be helpful to readers to end with a graphical illustration that summarize the findings.
- 3D is probably a more common and appropriate term than 'tridimensional' used by the authors

Reviewer #3 (Remarks to the Author):

The manuscript by Orré et al present a set of elegant data using advanced quantitative bioimaging approaches that unravelled the mobility behavior and nanoscale three-dimensional position of the protein kindlin-2 in the context of integrin activation at focal adhesions. Mobility within the plasma membrane as well as vertical positioning of kindlin molecules within single focal adhesions are important parameters to understand how kindlin immobilizes integrins and contribute to their activation in a manner different from talin. Since kindlin and talin are known to have a complementary action during integrin activation, this manuscript provides interesting biophysical insights into the role of kindlin: while talin comes to the FAs directly from the cytosol, kindlin is already diffusing within the plasma membrane via its PH domain before reaching the FAs.

The manuscript is very well-written, the figures are rich but very clear, the methodology is elegant and thorough and appeals to a broad readership and the statements provided are sufficiently novel and link mobility to nanoscale 3D localization of an important integrin activator. Therefore, in my view, this study deserves publication. However, there are a number of points the authors must first address in a revised manuscript.

1) the authors show that beta1 and beta3 integrins seem differently sensitive to impairment of the interaction with kindlin in Figure 1. It is not clear how useful and important these data are for the remaining of the story as they are not discussed further.

2) a number of kindlin-2 (K2) mutants are overexpressed in MEFs for the sptPALM experiments, I assume that with the exception of fig 7 and S7, where MEFs K1 and K2 KO are used, all other MEFs are wild-type, thus expressing kindlin WT. I understand the co-expression of fluorescent K2 mutants in normal MEFs allows to track the behavior of the mutated kindlin in the context of normally formed FAs. However, it would be very useful to relate the data on mobility and 3D location with data of FA features: are the number and morphology of FAs in MEFs overexpressing the K2 mutants (QW, L357A, K390A, DeltaPH) similar to wild-type MEFs? In Fig S8 important parameters are reported in Kind KO MEFs expressing K2-DeltaPH and K2-DeltaPH-CAAX mutants. These parameters should be provided also for the other mutants used. This would show the effect of altered mobility and/or 3D localization on FA properties.

3) could the authors explain why K2-DeltaPH is immobile outside FAs?

4) data in Fig 1e,f are also shown in Fig S1 e,f: is this necessary?

5) why is beta3 wild-type integrin immobile outside FAs (Fig 1h)? Why is talin so highly immobile outside FAs (Fig 2d)? Why is 70% of paxillin immobile outside FAs (Fig S5c)?

6) the Diff coefficients are calculated from the small fraction that is mobile. which is very small inside FAs, at the same time the changes in D values are also very very modest: they are statistically different, but what do these tiny changes tell biologically? Could the authors strengthen their interpretation of these data?

7) Figure 6: I appreciate the complexity of the imaging approaches used in this study but presenting data in a main figure that are obtained by two or one experiment is not sufficiently sound. In Fig. S6, four experiments are mentioned in which the same mutants used for figure 6 were used. I therefore assume the authors have the possibility to add more experimental data and strengthen the results shown in fig 6. In addition, I am wondering why the information provided in fig6 is actually not shown and discussed earlier in the manuscript, right before the authors embark on sptPALM experiments? What is the reason for not putting these data as supplementary figure?

8) the data provided in Figure 7 are used a conclusive statement to link the biophysical parameters to integrin activation (cell spreading). However, these data are not entirely novel as the eLife paper of the Faessler group already provided the same knowledge. I am therefore wondering: why not using these data as starting point? As motivation to better understand mechanistically what drives these differences? Mobility and nanoscale 3D localization of kindlin would then be the mechanistic explanation. In Fig S8, interesting data are shown for individual FAs formed after expression of two out of the five kindlin-2 mutants used. I think the same parameters should be provided for the QW, L357A and K390A mutants, to link the effects of these mutations to individual FA properties and eventually to cell spreading. Unfortunately, Fig S8 seems not mentioned in the Results section.

Alessandra Cambi

REVIEWER #1 (REMARKS TO THE AUTHOR):

In the present manuscript, T. Orré and colleagues use single particle tracking and 3D nanoscopy to address the localization and dynamics of kindlin 2 in focal adhesions. In particular, they reveal that kindlin is important for integrin immobilization at focal adhesions and that, contrarily to talin, it freely diffuses at the plasma membrane. Furthermore, they demonstrate the importance of its PH domain for its localization. This very complete, well-documented and clear manuscript constitutes an important source of information to understand the role of kindlin 2 in cell adhesion. However, some information could be added and clarified to strengthen the impact of the manuscript.

We are grateful to the reviewer for her/his positive evaluation of our work and constructive suggestions. Furthermore, we appreciate the thorough evaluation of the reviewer #1 and we took into account his/her advices very seriously to strengthen the impact of our manuscript.

In addition to the insightful questions and interesting experiments proposed by the reviewer#1, we also performed additional experiments, suggested by the other reviewers, to improve and strengthen our manuscript.

This led to 4 modified main figures (Fig. 2, 5, 6, 7) and 3 supplementary figures (Supplementary Fig. S3, S4, S9). Note that the figures and supplementary figures are now numbered according to the revised manuscript.

The main results are summarized below:

- As found in mature focal adhesions (FAs), kindlin-2 is immobile and enriched in early nascent adhesions (NAs) (Supplementary Fig. S3), and displays membrane free diffusion both inside and outside NAs (Supplementary Fig. S3). Similarly, a kindlin-2 mutant with impaired binding to integrins (kindlin-2-QW) displayed a decreased immobile fraction in NAs compared to kindlin-2 (Supplementary Fig. S3), suggesting that association between integrin and kindlin contributes to kindlin-2 immobilization in NAs. The results we found in NAs mirrors the ones found in mature FAs. Thus, the molecular mechanisms leading to integrin activation by kindlin-2 in FAs and NAs are probably closely related.

- To further link the molecular behavior of kindlin with its function in integrin activation inside FAs, we performed additional experiments concerning the formation of FAs in kindlin-1,2 Knock Out fibroblasts (Fig. 7). In the original version of the manuscript we performed rescue experiments in kindlin-1,2 KO MEFs, concerning FAs formation, for kindlin-2-WT, kindlin-2-ΔPH, and kindlin-2-ΔPH-CAAX. In the revised manuscript we performed additional experiments and quantifications for kindlin-2-QW and kindlin-2-CAAX. Importantly, these new results on FAs formation follow a similar trend to the one quantified for these mutants regarding enrichment in FAs (Fig. 6), and cell spreading (Fig. 7): Kindlin-2 = Kindlin-2-CAAX > Kindlin-2-ΔPH-CAAX = Kindlin-2-QW > Kindlin-2-ΔPH. As demonstrated in our study this trend in cell spreading and FAs formation reflects the ability of kindlin-2-WT and mutants to bind and activate integrins, which is driven by membrane recruitment, membrane free-diffusion, and 3D nanoscale localization in the integrin layer inside FAs.
- We performed additional experiments and quantification to increase the number of independent experiments concerning the enrichment in FAs of kindlin-2 mutants (Fig. 6). The results obtained confirmed what we have found in the original manuscript and follow the trend: Kindlin-2 = Kindlin-2-K390A > Kindlin-2-ΔPH-CAAX = Kindlin-2-L357A > Kindlin-2-QW > Kindlin-2-ΔPH > cytosolic mEos2.
- Kindlin-2 dwell-time and rearward motion in mature FAs. Finally, we generated super-resolved time-lapses and kymographs (Fig. 2) to measure kindlin-2 dwell-time in mature FAs and to test whether kindlin-2 is moving rearward inside FAs as demonstrated for talin. First, we found that immobilization durations were much shorter for kindlin-2 compared to talin-1 (Fig. 2h, i). Analysis of longer kymographs showed that the fraction of immobilized kindlin-2 undergoing retrograde flow with speed above 2 nm.s⁻¹ was around 40 % similar to what was measured for talin-1 (Fig. 2j, k)(Rossier et al., 2012). Our results suggest that the interaction of kindlin-2 with stationary integrins is more prevalent than with F-actin, because the fraction of F-actin moving rearward above 2 nm.s⁻¹ is around 75 %(Rossier et al., 2012). We added these new results in Fig. 2, and we mentioned these results in the results section (p.6) and the discussion section to establish a model with the sequential formation of a transient tripartite integrin-kindlin-talin complex (p.12).
- We have performed all the additional quantifications that has been proposed by the reviewers (Fig. 2, 5, 6, 7 and Supplementary Fig. S3, S4, S9).

Main comments:

-The authors chose to explore the function of kindlin in mature focal adhesions only, and not in nascent adhesions. However, it is not specified how mature adhesions, and not nascent adhesions, were selected for the analysis. Furthermore, as kindlin was shown to be important for the formation of nascent adhesions (Bachir et al. 2014; Theodosiou et al. 2016; Böttcher et al. 2017), it would also be interesting to address the dynamics of kindlin in nascent adhesions.

In mesenchymal cells such as fibroblasts, nascent adhesions (NAs) form in protrusive structures such as the lamellipodium. We studied extensively the formation of nascent/early adhesions in the lamellipodium of fibroblasts (Giannone et al., Cell 2004, Cell 2007; Dubin-Thaler et al., Biophysical Journal 2004). In those articles, we revealed protrusion/retraction cycles ('periodic contractions') during lamellipodium protrusions during cell spreading but also cell migration. We used the same Mouse Embryonic Fibroblasts (MEFs) in these published articles and in the current submitted manuscript.

In these articles, we performed a precise characterization of lamellipodium dynamics but also of integrin, actin and myosin II regulators involved in this phenomenon (VASP, integrin, paxillin, myosin light chain, MLCK, α -actinin). Thus we know where are located all these proteins with the classical optical resolution of ~ 250 nm. In these previous studies and in the current submitted manuscript, we can clearly discriminate NAs that are initiated $0.5 \mu\text{m}$ back from the lamellipodium tip (Giannone et al., Cell 2004, Cell 2007) and mature focal adhesions (FAs) that result from the subsequent maturation of a fraction of NAs outside of the lamellipodium. In addition, in all the experiments we performed we used MEFs spread on fibronectin for more than 3 hours possessing well-defined mature FAs. Region of interests where chosen outside of active lamellipodia where are located NAs.

We added 3 sentences in the method section to clarify this point (p. 15):

“Cells were co-transfected with mEos2-fused proteins and GFP-paxillin as a FA reporter. To clearly discriminate NAs, initiated $0.5 \mu\text{m}$ back from the lamellipodium tip (Giannone et al., Cell 2004, Cell 2007) from mature FAs, resulting from the subsequent maturation of a fraction of NAs outside of the lamellipodium, we used MEFs spread on fibronectin for more than 3 hours possessing well-defined mature FAs. Region of interests where chosen outside of active lamellipodia where NAs are located. For results obtained in early NAs (Supplementary Fig. S3), we performed experiments in active lamellipodia during cell spreading between 15 to 60 min after cell loading (Giannone et al., Cell 2004, Cell 2007).”

Following the advice of reviewer#1, in the revised version of the manuscript, we studied the diffusive behavior of kindlin-2 in NAs of MEFs. We performed sptPALM experiments on NAs in lamellipodia of spreading MEFs using mEos2-kindlin-2 and mEos2-kindlin-2-QW614/615AA (impaired interaction with integrins). As found in mature FAs, mEos2-kindlin-2 is immobile and enriched in NAs (Supplementary Fig. S3), and displays membrane free diffusion both inside and outside NAs (Supplementary Fig. S3). As in FAs, kindlin-2-QW displayed a decreased immobile fraction compared to kindlin-2-WT (Supplementary Fig. S3), suggesting that association between integrin and kindlin contributes to kindlin-2 immobilization in NAs. The results we found in NAs mirrored those found in FAs. Thus, the molecular mechanisms leading to integrin activation by kindlin-2 in FAs and NAs are probably closely related.

Those new results about kindlin-2 in NAs were added in the results section (p. 6) in “Kindlin-2 undergoes free diffusion along the plasma membrane, inside and outside FAs”:

“Furthermore, as kindlin was shown to be important for the formation of NAs (Bachir et al. 2014; Theodosiou et al. 2016; Böttcher et al. 2017), we also studied the diffusive behavior of kindlin-2 in NAs within the lamellipodia of spreading MEFs. Like found in mature FAs, mEos2-kindlin-2 is immobile and enriched in NAs (Supplementary Fig. S3), and displays membrane free diffusion both inside and outside NAs (Supplementary Fig. S3). Altogether, these results indicate that membrane free diffusion is a general feature of the kindlin family that could confer distinct capabilities compared to talin during integrin activation in NAs and FAs”

and in “Disruption of the integrin-kindlin interaction increases kindlin diffusion inside FAs”:

“We obtained the same results for kindlin-2-QW in NAs (Supplementary Fig. S3). The remaining fraction of immobilized kindlin-2-QW could result from binding to other partners, especially...”

-Fig. 4d shows that the PH domain alone displays the same fraction of immobilization than K2-WT. This seems inconsistent with Fig. 3 that would argue for a dependency on interaction with integrin for kindlin immobilization. This should be commented.

We would like to thank the reviewer for the detailed evaluation of our study. It is true that a decreased immobilization fraction in conjunction with an increased fraction of membrane free-diffusion often reflects a shift in population from an immobile pool engaged with a binding partner inside FAs to a freely diffusing pool detached from this binding partner. This is the case for instance for kindlin-2 versus kindlin-2-QW (Fig. 3 in MEFs and Supplementary Fig. S8 in Kind^{Ko} cell), or β 31-integrin versus β 31-integrin-Y795A (Fig. 1).

However, we want to emphasize that the immobilization fraction is not always correlated with the immobilization density or recruitment within a subcellular structure. For example, it is possible to obtain similar immobilization fractions but very different immobilization densities. For the same area, you can find 10 stationary and 10 free-diffusive trajectories, or 100 stationary and 100 free-diffusive trajectories, the immobilization fraction (50%) will be the same but the density will be an order of magnitude different. This is illustrated by the diffusive behavior of talin-1, which exhibits similar levels of immobilization inside compared to outside FAs (82% versus 73%, Fig. 2d), but with a much higher immobilization density inside than outside FAs (~5 times more detections inside than outside as shown in Rossier et al., Nature Cell Biology 2012 - Fig. 4f). This is also the case, for instance, for paxillin, which possesses large immobile fractions inside and outside FAs but is specifically enriched in FAs (Supplementary Fig. S6). Likewise, kindlin-2 and kindlin-2- Δ PH exhibit similar immobile fractions within FAs (Fig. 4) despite the reduced enrichment of kindlin-2- Δ PH in FAs compared to kindlin-2 (Fig. 6f).

As pointed out by the reviewer, this is also the case for the PH-domain of kindlin-2 that display similar immobile fraction within FAs than kindlin-2. However, its recruitment in FAs is reduced, as showed by the quantification of the density of detections inside vs. outside FAs for kindlin-2, and for the PH domain of kindlin-2 (kindlin-2: 5.7 times more detections inside than outside; PH-domain: 2.7 times more detections inside than outside). Thus, the PH domain of kindlin-2 within FAs is much less prone to immobilization than kindlin-2. Nevertheless, these immobilizations of the PH-domain of kindlin-2 inside FAs could be due to a direct interaction with paxillin, as previously described (Theodosiou eLife 2016), or to other unknown binding partners.

-In Fig.5, the authors detail the nanoscale 3D localization of kindlin in focal adhesions, taking as references paxillin. First, there is a small discrepancy in the axial localization of paxillin compared with the results published by Kanchanawong et al. (Nature 2010). Second, according to Kanchanawong et al., Kindlin being located at 48.5 nm above the coverslip, it should be considered in the “force transduction layer”, rather than in the integrin layer. Since there seems to be some differences with the work of Kanchanawong et al., I suggest to determine the axial localization of other markers of the different layers of focal adhesions (e.g. CAAX, integrins, vinculin and actinin) to be able to conclude about the layer in which kindlin is localized.

The reviewer is correct in saying that the axial localizations found using our experimental workflow are shifted upwards compared to the axial localizations found in the study published by Kanchanawong et al. Nature 2010. However, we think that we can explain this apparent discrepancy.

Indeed, the axial localization found for GFP-kindlin-2 in our study was $Z_{\text{peak}} = 48.7$ nm. The localization we found for paxillin-Nterm was $Z_{\text{peak}} = 58.9$ nm, compared to $Z_{\text{center}} = 46.2$ nm found in the study published by Kanchanawong et al. (Nature 2010). Note that in the same study, the axial localization of paxillin-Cterm ($Z_{\text{center}} = 43.1$) is below the one of paxillin-Nter ($Z_{\text{center}} = 46.2$ nm), highlighting that the axial localization also depends on the orientation of proteins, as found for talin (Kanchanawong et al., Nature 2010) and vinculin (Case et al., Nature Cell Biology 2015). Thus, the localization we obtained for paxillin-Nterm is shifted upwards by 12.7 nm (58.9 nm - 46.2 nm) from that found in the Kanchanawong article. It should be highlighted that the axial localization found for CAAX-tdEos in Kanchanawong's is $Z_{\text{center}} = 32.3$ nm, while we found an axial localization of 39.0 nm for GFP-kindlin-2- Δ PH-CAAX, which could also be considered as a close marker of the plasma membrane. Importantly, we again obtained a similar upward shift of few nanometers, 6.7 nm (39.0 nm - 32.3 nm). Thus it seems that the axial localizations found in our study are upwardly shifted of few nanometers compared to the ones found in Kanchanawong's study.

It is important to note that in the study published by Kanchanawong et al. (Nature 2010), the authors used iPALM to localize the Z position of proteins labelled with a photo-convertible fluorescent proteins (tdEos or mEos2). Thus they measure the axial localization of photo-convertible fluorescent proteins (PA-FPs): target protein + **PA-FPs**. In our study, we used GFP-tagged proteins that we imaged by dSTORM using anti-GFP nanobodies labelled with AlexaFluor-647. Thus we measure the axial localization of the AlexaFluor: target protein + GFP + Nanobody + **AlexaFluor**. To note, the size of a GFP is around 4.2 nm x 2.4 nm, while the size of a Nanobody is around 4.8 nm x 2.2 nm. Therefore, the shifted axial localization we obtained could be in part explained by the additional layers/components we used to label the protein of interest.

In the study of Kanchanawong, paxillin is defined as being part of the integrin signaling layer. Thus, in any case, we can use GFP-paxillin-Nterm as the upper limit of the integrin signaling layer. Since, in our DONALD experiments, the axial localization of GFP-kindlin-2 (Nterm labelled, $Z_{\text{peak}} = 48.7$ nm) is below the one of GFP-paxillin (Nterm labelled, $Z_{\text{peak}} = 58.9$ nm), we think that we can also define Kindlin-2 as being part of the integrin signaling layer.

-It is not clear whether the 3D nanoscopy analysis was performed only in focal adhesions or in the whole cell edges. As for the other figures, I also suggest to perform these measurements both inside and outside adhesions.

In the initial submission, the results displayed in Fig. 5 related to 3D nanoscopy correspond to axial localizations measured within mature focal adhesions. Following the advice of the reviewer, we also analyzed the axial localizations outside focal adhesions. These results are now presented in Fig. 5 together with the axial localisations inside FAs. The results show that compared to inside FAs, proteins' axial localization outside FAs is generally shifted upwards but conserves the same trends between kindlin-2 and mutants with: kindlin-2- Δ PH-CAAX < kindlin-2 = kindlin-2-QW < kindlin-2- Δ PH. Paxillin displays the widest shift in amplitude between inside and outside FAs. As paxillin outside FAs display no membrane free-diffusion and membrane recruitment according to sptPALM results (Supplementary Fig. S6), paxillin axial localization outside FAs corresponds to the distribution of a cytosolic protein in our experimental conditions. Kindlin-2- Δ PH axial localization is close to the one of paxillin ($Z_{\text{peak}} = 81.9$ nm) whereas kindlin-2 ($Z_{\text{peak}} = 69.2$ nm) and kindlin-2-QW ($Z_{\text{peak}} = 67.8$ nm) are maintaining a close proximity (<10 nm) to the plasma membrane as localized by the axial localization of kindlin-2- Δ PH-CAAX ($Z_{\text{peak}} = 60.7$

nm). This further reinforces the idea that the kindlin-2 PH domain is an important determinant to maintain kindlin-2 at the plasma membrane even outside FAs.

Minor comments:

-The introduction of this manuscript lacks an introduction on the current knowledge on the function of kindlins at focal adhesions.

We added a paragraph to introduce the current knowledge about kindlin functions in integrin adhesions and its potential interplay with talin in p. 3:

“Talin, the most described integrin activator, is recruited inside FAs directly from the cytosol (Rossier et al., 2012) and controls integrin immobilization by binding to the proximal NPxY motif on integrin β cytoplasmic tail with its FERM domain (4.1-protein/ezrin/radixin/moesin) (Calderwood et al., 2013). More recently, kindlins were demonstrated to be critical for integrin activation (Bouaouina et al., 2012; Goult et al., 2009; Shi et al., 2007). Kindlins also possess a FERM domain which binds to β -integrin via the distal NxxY motif (Montanez et al. 2008). Integrin activation by kindlins has been predominantly described during cell spreading and the formation of nascent adhesions (NAs) within active protrusions of the cell (Bachir et al., 2014; Böttcher et al., 2017; Theodosiou et al., 2016) rather than in mature FAs. Nevertheless, talins and kindlins were demonstrated to cooperate during integrin activation (Moser et al. 2009; Theodosiou et al. 2016; Legate & Fassler 2009). Indeed, kindlin is often required for proper integrin activation by talin (Huet-Calderwood et al., 2014; Lai-Cheong et al., 2009; Montanez et al., 2008; Moser et al., 2008; Qu et al., 2011; Ruppert et al., 2015; Theodosiou et al., 2016; Ussar et al., 2008). Moreover, kindlin and talin activate integrin not simply in an additive manner, but rather in a synergistic way (Ma et al., 2008; Montanez et al., 2008; Theodosiou et al., 2016; Ussar et al., 2008). Thus, kindlins and talins may cooperate to activate integrins by binding distinct regions of the β integrin tail; however, the molecular events underlying this cooperation are poorly understood (Moser et al., 2009).”

-It should be indicated that beta3-Y747A is not specific of talin binding as it also decreases the interaction with other proteins, including FlnA (Petrich et al. 2007).

We added a sentence in the results section (p. 5) indicating that this point mutation also affects binding to filaminA and also tensin:

“Consistent with our previous results (Rossier et al., 2012), a mutation in the membrane-proximal NPxY (81-Y783A (Anthis et al., 2010) (Anthis et al., 2010) (Anthis et al. 2010 Structure (London, England: 1993)) (Anthis et al., 2010) or 83-Y747A (Tadokoro et al., 2003)), which was shown to decrease the interaction with talin, filamin and tensin (Tadokoro et al. 2003), ...’.”

-I propose to discuss the possible dimerization of kindlin, and in particular how it could affect kindlin diffusion.

We thank the reviewer for this interesting suggestion. As we obtained by sptPALM similar results for kindlin-2 and several mutants in absence or presence of endogenous kindlin-2 (respectively with Kind^{KO} cells and wild-type MEFs), we decided to mention this possibility in the results section (p. 8) as followed:

“Furthermore, the similarity of diffusive behavior found in wild-type MEFs and Kind^{Ko} cells suggests that sptPALM results obtained in WT MEFs are not biased by potential formation of heterodimers between endogenous kindlins and transfected kindlin-2 mutants as suggested by structural studies (Li et al., 2017).”

-Result/Disruption of the integrin-kindlin interaction increases kindlin diffusion inside FAs: “These results support the idea that kindlin controls integrin immobilization inside mature FAs.” Shouldn’t it be written: “These results support the idea that integrin controls kindlin immobilization inside mature FAs.” ?

We thank the reviewer for this correction. We changed the sentence accordingly on p.6 as followed:

“Thus, our results support the idea that integrin are playing a crucial role in kindlin immobilization inside mature FAs.”

-O. Rossier et al. have previously shown that talin is partly moving rearward in focal adhesions. It would be interesting to address whether it is also the case for kindlin, in order to evaluate to what extent it is coupled to the actin retrograde flow.

We thank the reviewer for this question and comments. As asked by the reviewer, we tested whether kindlin-2 is moving along the actin retrograde flow in mature focal adhesions (FAs) as demonstrated for talin. We generated super-resolved time-lapses and kymographs (Fig. 2) to measure kindlin-2 rearward flow in FAs, but also kindlin-2 immobilization durations (as done for β -integrins in the original version of the manuscript, Fig. 1i-l).

First, the durations of kindlin-2 immobilizations were much shorter than those measured for talin (Fig. 2). As kindlin-2 and talin-1 immobilizations in FAs correspond in part to interactions with immobilized integrins, these results suggest that kindlin-integrin interactions are shorter than talin-integrin interactions. These results suggest a low occurrence of stable tripartite integrin-kindlin-talin complexes, otherwise the immobilization durations would have been similar for kindlin and talin. These results rather support the existence of transient tripartite integrin-kindlin-talin complex, either enabling the initiation of integrin-talin interactions, or extending the duration of integrin/talin interactions.

Second, analysis of kymographs showed that the fraction of kindlin-2 undergoing retrograde flow with speed above $2 \text{ nm}\cdot\text{s}^{-1}$ was 40 % which is similar to what was measured for talin-1 (50 % in Fig. 2). In comparison, the fraction of actin undergoing retrograde flow above $2 \text{ nm}\cdot\text{s}^{-1}$ is around 75 %. These results rather suggest that kindlin-2 is not directly and only bound to actin in mature FAs, otherwise we would have obtained a larger fraction of retrograde flow in FAs. These, results rather suggest, like for talin, that kindlin-2 in FAs are mainly interacting with stationary integrins. The fraction of kindlin-2 undergoing retrograde flow could correspond to kindlin-2 interacting with the small fraction of flowing $\beta 1$ -integrin (Rossier et al., NCB 2012), or a small fraction of kindlin-2 binding directly or indirectly to actin filaments (Bledzka JCB 2016) without being interacting with stationary integrins. We added these new results in Fig. 2 and in the results section (p.6):

“Interestingly, using super-resolved time-lapses and kymographs (Fig. 2f-h) we found that immobilization durations were much shorter for kindlin-2 compared to talin-1 (Fig. 2i). Analysis of longer kymographs showed that the fraction of immobilized kindlin-2 undergoing retrograde flow

with speed above 2 nm.s⁻¹ was around 40 % similar to what was measured for talin-1 (Fig. 2j, k)(Rossier NCB 2012). Our results suggest that the interaction of kindlin-2 with stationary integrins is more prevalent than with F-actin, because the fraction of F-actin moving rearward above 2 nm.s⁻¹ is around 75 % (Rossier NCB 2012)."

and we used these results in the discussion section to establish a model with the sequential formation of a transient tripartite integrin-kindlin-talin complex (p. 12; Fig. 7) as follows:

"As the probability for three proteins to meet simultaneously with the correct orientation is extremely low, we favor a model with the sequential formation of an immobile integrin-kindlin complex followed by the formation of a transient tripartite integrin-kindlin-talin complex where kindlin could be replaced by talin head and reciprocally (Fig. 7). Indeed, as kindlin-2 and talin-1 immobilizations in FAs correspond in part to interactions with immobilized integrins, the shorter duration of kindlin-2 immobilizations suggests that kindlin-integrin interactions are shorter than talin-integrin interactions. This also suggests a low occurrence of stable tripartite integrin-kindlin-talin complexes, otherwise the immobilization durations would have been similar for kindlin and talin. These results rather support the existence of transient tripartite integrin-kindlin-talin complex, either enabling the initiation of integrin-talin interactions, or extending the duration of integrin/talin interactions. The immobile integrin-kindlin complex could constrain the integrin β -tail orientation and favor the binding of talin to the proximal NPxY motif."

-Fig.5h: I suggest to represent data similarly to Fig.4a-c from Kanchanawong et al., and to display orthogonal sections of FAs.

This would be a good method to represent the data, but we do not currently have a readily available approach to achieve this type of representation.

-The authors propose in the discussion that kindlin could stabilize integrin-talin interactions. This is an important question that could be addressed by determining the effect of kindlin KO on talin diffusion.

This is a very interesting suggestion. However, it is impossible to obtain kindlin KO cells that will form mature FAs. Indeed, kindlin-1,2 KO cells do not spread unless the cells are treated with Mn²⁺ to activate integrins. Even under these conditions, the cells will not form mature FAs but only nascent adhesions (Theodosiou et al., eLife 2016). One alternative way to answer this question would be perhaps to perform dual color Single Protein Tracking to track simultaneously kindlin and talin and study the interactions between these two proteins. We are planning to perform such type of experiments in future studies.

-It would help the reader to add a cartoon that summarizes the findings of this study.

We agree with the reviewer, we have added a cartoon summarizing our findings in Fig. 7.

REVIEWER #2 (REMARKS TO THE AUTHOR):

In this study, live-cell single-particle tracking (SPT) and super-resolution microscopy techniques were applied to study the motion characteristics of kindlin in relation to its localization to focal adhesions (FA). Based on site-specific mutations, integrin- β 1 and β 3 immobilization in FAs is shown to be dependent on the binding of their cytoplasmic domains to kindlin. Kindlin-1 and -2 were shown to undergo primarily lateral diffusion in the membrane plane, in contrast to Talin which enter FA largely via 3D diffusion. Kindlin-2 immobilization in FA depends on its interaction with activated integrin, but not paxillin or ILK. Kindlin-2 membrane diffusion is mediated by PH domain and interaction with phosphoinositides, and is important for nanoscale targeting to the membrane-proximal compartment. Kindlin association with the membrane is shown to be important for FA localization and enrichment, and promote cell spreading.

Overall this is a rigorous study using state-of-the-art techniques and KO cell lines established earlier by the authors. The approach gives mechanistic insight that is highly informative on the function of kindlin in cell adhesions. Data is well-presented and the experiments are well-controlled. I am supportive of its eventual acceptance to Nature Communications. However there is one important point that should be addressed.

We are grateful to the reviewer for her/his positive evaluation of our work and constructive suggestions. Furthermore, we appreciate the thorough evaluation of the reviewer #2 and we took into account his/her advices very seriously to strengthen the impact of our manuscript.

In addition to the insightful questions and interesting experiments proposed by the reviewer#2, we also performed additional experiments, suggested by the other reviewers, to improve and strengthen our manuscript.

This led to 4 modified main figures (Fig. 2, 5, 6, 7) and 3 supplementary figures (Supplementary Fig. S3, S4, S9). Note that the figures and supplementary figures are now numbered according to the revised manuscript.

The main results are summarized below:

- In the results section of the revised manuscript, we comment further the results obtained with the ILK binding defective kindlin-2 mutant (kindlin-2-L357A). These comments are based on results obtained in the article of Huet-Calderwood et al, JCS 2014, and on additional western blots we performed to compare the expression levels of endogenous kindlin-2 in the MEFs we used, but also in CHO cells with expression levels of transfected kindlin-2 or kindlin-2-L357A in MEFs. Those new results are displayed in Supplementary Fig. S4.
- As found in mature focal adhesions (FAs), kindlin-2 is immobile and enriched in early nascent adhesions (NAs) (Supplementary Fig. S3), and displays membrane free diffusion both inside and outside NAs (Supplementary Fig. S3). Similarly, a kindlin-2 mutant with impaired binding to integrins (kindlin-2-QW) displayed a decreased immobile fraction in NAs compared to kindlin-2 (Supplementary Fig. S3), suggesting that association between integrin and kindlin contributes to kindlin-2 immobilization in NAs. The results we found in NAs mirrors the ones found in mature FAs. Thus, the molecular mechanisms leading to integrin activation by kindlin-2 in FAs and NAs are probably closely related.
- To further link the molecular behavior of kindlin with its function in integrin activation inside FAs, we performed additional experiments concerning the formation of FAs in kindlin-1,2 Knock Out fibroblasts (modified Fig. 7). In the original version of the manuscript we performed rescue

experiments in kindlin-1,2 KO MEFs, concerning FAs formation, for kindlin-2-WT, kindlin-2- Δ PH, and kindlin-2- Δ PH-CAAX. In the revised manuscript we performed additional experiments and quantifications for kindlin-2-QW and kindlin-2-CAAX. Importantly, these new results on FAs formation follow a similar trend to the one quantified for these mutants regarding enrichment in FAs (Fig. 6), and cell spreading (Fig. 7): Kindlin-2 = Kindlin-2-CAAX > Kindlin-2- Δ PH-CAAX = Kindlin-2-QW > Kindlin-2- Δ PH. As demonstrated in our study this trend in cell spreading and FAs formation reflects the ability of kindlin-2-WT and mutants to bind and activate integrins, which is driven by membrane recruitment, membrane free-diffusion, and 3D nanoscale localization in the integrin layer inside FAs.

- We performed additional experiments and quantification to increase the number of independent experiments concerning the enrichment in FAs of kindlin-2 mutants (modified Fig. 6). The results obtained confirmed what we have found in the original manuscript and follow the trend: Kindlin-2 = Kindlin-2-K390A > Kindlin-2- Δ PH-CAAX = Kindlin-2-L357A > Kindlin-2-QW > Kindlin-2- Δ PH > cytosolic mEos2.
- Kindlin-2 dwell-time and rearward motion in mature FAs. Finally, we generated super-resolved time-lapses and kymographs (Fig. 2) to measure kindlin-2 dwell-time in mature FAs and to test whether kindlin-2 is moving rearward inside FAs as demonstrated for talin. First, we found that immobilization durations were much shorter for kindlin-2 compared to talin-1 (Fig. 2h, i). Analysis of longer kymographs showed that the fraction of immobilized kindlin-2 undergoing retrograde flow with speed above 2 nm.s⁻¹ was around 40 % similar to what was measured for talin-1 (Fig. 2j, k)(Rossier et al., 2012). Our results suggest that the interaction of kindlin-2 with stationary integrins is more prevalent than with F-actin, because the fraction of F-actin moving rearward above 2 nm.s⁻¹ is around 75 %(Rossier et al., 2012). We added these new results in Fig. 2, and we mentioned these results in the results section (p.6) and the discussion section to establish a model with the sequential formation of a transient tripartite integrin-kindlin-talin complex (p.12).
- We have performed all the additional quantifications that has been proposed by the reviewers (Fig. 2, 5, 6, 7 and Supplementary Fig. S3, S4, S9).

1. The Calderwood lab showed in 2014 (Huet-Calderwood et al. J Cell Sci) that ILK and Kindlin-2 interactions appeared to be important for kindlin targeting to FAs. However, this seems to contrast with the results in Fig. S3 in this study where the ILK-binding mutant of kindlin, L357A shows essentially the same behaviors as WT. In the Calderwood study, L357A kindlin-2 seems to have FA-localization defect. However, in inset of Fig. S3a, the FA localization of L357A kindlin-2 seems to be quite substantial. While the authors may have mentioned these localization differences in passing, given the extensive and rigorous works on ILK, Kindlin-2, and FAs reported earlier by the Calderwood lab, I feel that any differences on ILK/Kindlin-2 interactions observed here should be looked into or discussed at some depth in context of these earlier findings.

We thank the reviewer for this wise comment and for pointing out those differences.

In the article of Huet-Calderwood et al, JCS 2014 it was demonstrated that kindlin-2 recruitment in FAs depends on an interaction with ILK. Indeed, a point mutation in kindlin-2 (L357A) inhibits the interaction between ILK and kindlin-2 in biochemistry pull-down assays (Fig. 6D in cited reference) and also inhibits kindlin-2 recruitment in FAs, quantified using epifluorescence (Fig. 7A in cited reference). Those

experiments were mainly performed in CHO cells. Note that in the Huet-Calderwood article, the authors also found that kindlin-2 could be weakly recruited in FAs in ILK Knock Out fibroblasts (Fig. 7C in cited reference). They also observed, in contrast with what they obtained in CHO cells, that GFP-kindlin-2-L357A displayed recruitment to FAs in ILK KO similar to that of GFP-kindlin-2 (Fig. 7D in cited reference). These results suggest that kindlin-2 could also be recruited in FAs independently from its binding to ILK.

Importantly, the authors also found that there could be competition between different kindlin isoforms for their recruitment inside FAs. For instance, kindlin-3, the hematopoietic-specific kindlin, is not strongly recruited in FAs in fibroblast since it might be outcompeted by endogenous kindlin-2. However, in kindlin-2 Knock Down fibroblasts, the authors could observe recruitment of kindlin-3-GFP in FAs (Fig. 8C in cited reference). Furthermore, in Kindlin-2 Knock Down fibroblasts GFP-kindlin-2-L357A is also significantly recruited in FAs. In fact, the authors also observed a reduced expression of kindlin-2 in ILK KO, which might explain why GFP-kindlin-2-L357A could be weakly recruited in FAs in these cells. Altogether, these results suggest that ILK binding is not indispensable for FAs targeting in the absence of competing endogenous kindlin-2. These results also show that kindlin-2 could be recruited in FAs independently from its binding to ILK, as explained by the authors in their article.

In our sptPALM experiments the ILK binding defective kindlin-2 mutant (mEos2-Kindlin-2-L357A) behaves similarly to mEos2-Kindlin-2, both in WT Mouse Embryonic Fibroblasts (MEFs) (Supplementary Fig. S4) and in Kindlin-1,2 KO MEFs (Supplementary Fig. S8h,i). Furthermore, compared to kindlin-2 mutants used in our study (mEos2-kindlin-2-QW, mEos2-kindlin-2- Δ PH), mEos2-kindlin-2-L357A is enriched in mature FAs to almost a similar level, albeit slightly less enriched than mEos2-kindlin-2 (Fig. 6).

Based on the results obtained in Huet-Calderwood et al, JCS 2014, one possible explanation is that endogenous kindlin-2 expression level is low in the WT MEFs we used in our study compared to CHO cells. If this is the case, expression of exogenous kindlin-2-L357A could outcompete endogenous kindlin-2, as proposed by Huet-Calderwood and colleagues for ILK KO and in Kindlin-2 Knock Down fibroblasts. To test this hypothesis, we performed western blots to compare the expression levels of endogenous kindlin-2 in the MEFs we used and in CHO cells. However, the expression level of endogenous kindlin-2 in our MEFs was about 2 fold the level found in CHO. Thus similar FAs recruitment and diffusive behavior found for kindlin-2-L357A and kindlin-2-WT in our experiments could not be explained solely by a low expression of endogenous kindlin-2.

Transfected kindlin-2-L357A might also outcompete endogenous kindlin-2. We also performed western blots to compare the expression levels of endogenous kindlin-2 and transfected exogenous mEos2-kindlin-2 and mEos2-kindlin-2-L357A in MEFs. We used exactly the same condition of transfection than the one used for sptPALM experiments and FAs recruitment experiments. We found a 1.5-fold higher expression of transfected mEos2-kindlin-2 and mEos2-kindlin-2-L357A compared to the endogenous kindlin-2. The higher level of mEos2-kindlin-2-L357A compared to endogenous kindlin-2 might thus partly explain why kindlin-2-L357A behaves similarly than kindlin-2. However, the level of overexpression of exogenous mEos2-kindlin-2-L357A is not much higher than that of endogenous kindlin-2, so the transfected mEos2-kindlin-2-L357A could not compete with endogenous kindlin-2 to a level that accounts for the small effects found in our experiments.

If ILK binding was indispensable for kindlin-2 recruitment in FAs and immobilization in FAs, we would have expected stronger effects in sptPALM experiments and FAs recruitment experiments with mEos2-kindlin-2-L357A as found for example for the kindlin-2 mutant with decreased binding to integrins kindlin-2-QW

(Fig. 3, 6 and Supplementary Fig. S8f,g). Thus as the authors of the Huet-Calderwood's article explained in their discussion, the recruitment and function of kindlin-2 in FAs could also be independent from ILK binding, for instance via integrins, actin (Bledzka JCB 2016), paxillin (Theodosiou eLife 2016) or other unknown binding partners. Furthermore, the ILK-dependent recruitment of kindlin-2 in FAs seems to depend on the cellular context. Here is a passage taken from the discussion of the Huet-Calderwood's article commenting these seemingly contradictory results:

“Nonetheless, at least in some cells, ILK binding is not absolutely required and the residual FA localization of kindlin-2 in ILK-knockout cells is likely due to the integrin binding site in the F3 subdomain. Indeed, in the absence of competing endogenous kindlin-2, ILK-binding-defective kindlins can target to FAs, providing that their integrin-binding site is intact.”

In the results section (p. 6) of the our revised manuscript, we now comment further the results obtained with the ILK binding defective kindlin-2 mutant kindlin-2-L357A as followed:

“The remaining fraction of immobilized kindlin-2-QW could result from binding to other partners, especially paxillin(Böttcher et al., 2017), ILK(Huet-Calderwood et al., 2014) or from residual binding of kindlin-2-QW to integrin. However, a point mutation decreasing kindlin-2 binding to ILK (kindlin-2-L357A)(Huet-Calderwood et al., 2014) had no effects on kindlin-2 immobilizations in FAs (Supplementary Fig. S4). These immobilizations could result from ILK-independent recruitment of kindlin-2 inside FAs, as described previously in ILK deficient fibroblasts (Huet-Calderwood, JCS 2014), potentially by binding to integrins, paxillin (Theodosiou eLife 2016), or other unknown partners. Alternatively, transfected kindlin-2-L357A mutants can outcompete endogenous kindlin-2 in FAs, as showed in kindlin-2 knock down fibroblasts (Huet-Calderwood, JCS 2014). However, in our experimental conditions, the level of mEos2-Kindlin-2-L357A expression was 1.5 fold the one of endogenous kindlin-2 (Supplementary Fig. S4), which indicates that in our MEFs ILK-independent recruitment of kindlin is predominant. Thus, our results support the idea that integrin are playing a crucial role in kindlin immobilization inside mature FAs.”

Minor points:

- It would be helpful to readers to end with a graphical illustration that summarize the findings.

We agree with the reviewer and have added a graphical illustration summarizing our findings in Fig. 7.

- 3D is probably a more common and appropriate term that 'tridimensional' used by the authors We replaced “tridimensional” by 3D trough out the manuscript, but not in the title.

REVIEWER #3 (REMARKS TO THE AUTHOR):

The manuscript by Orré et al present a set of elegant data using advanced quantitative bioimaging approaches that unravelled the mobility behavior and nanoscale three-dimensional position of the protein kindlin-2 in the context of integrin activation at focal adhesions. Mobility within the plasma membrane as well as vertical positioning of kindlin molecules within single focal adhesions are important parameters to understand how kindlin immobilizes integrins and contribute to their activation in a manner different from talin. Since kindlin and talin are known to have a complementary action during integrin activation, this manuscript provides interesting biophysical insights into the role of kindlin: while talin comes to the FAs directly from the cytosol, kindlin is already diffusing within the plasma membrane via its PH domain before reaching the FAs.

The manuscript is very well-written, the figures are rich but very clear, the methodology is elegant and thorough and appeals to a broad readership and the statements provided are sufficiently novel and link mobility to nanoscale 3D localization of an important integrin activator. Therefore, in my view, this study deserves publication. However, there are a number of points the authors must first address in a revised manuscript.

We are grateful to the reviewer #3 (Alessandra Cambi) for her positive evaluation of our work and constructive suggestions. Furthermore, we appreciate the thorough evaluation of Alessandra Cambi and we took into account her advices very seriously to strengthen the impact of our manuscript.

In addition to the insightful questions and interesting experiments proposed by Alessandra Cambi, we also performed additional experiments, suggested by the other reviewers, to improve and strengthen our manuscript.

This led to 4 modified main figures (Fig. 2, 5, 6, 7) and 3 supplementary figures (Supplementary Fig. S3, S4, S9). Note that the figures and supplementary figures are now numbered according to the revised manuscript.

The main results are summarized below:

- As found in mature focal adhesions (FAs), kindlin-2 is immobile and enriched in early nascent adhesions (NAs) (Supplementary Fig. S3), and displays membrane free diffusion both inside and outside NAs (Supplementary Fig. S3). Similarly, a kindlin-2 mutant with impaired binding to integrins (kindlin-2-QW) displayed a decreased immobile fraction in NAs compared to kindlin-2 (Supplementary Fig. S3), suggesting that association between integrin and kindlin contributes to kindlin-2 immobilization in NAs. The results we found in NAs mirrors the ones found in mature FAs. Thus, the molecular mechanisms leading to integrin activation by kindlin-2 in FAs and NAs are probably closely related.
- To further link the molecular behavior of kindlin with its function in integrin activation inside FAs, we performed additional experiments concerning the formation of FAs in kindlin-1,2 Knock Out fibroblasts (modified Fig. 7). In the original version of the manuscript we performed rescue experiments in kindlin-1,2 KO MEFs, concerning FAs formation, for kindlin-2-WT, kindlin-2- Δ PH, and kindlin-2- Δ PH-CAAX. In the revised manuscript we performed additional experiments and quantifications for kindlin-2-QW and kindlin-2-CAAX. Importantly, these new results on FAs formation follow a similar trend to the one quantified for these mutants regarding enrichment in FAs (Fig. 6), and cell spreading (Fig. 7): Kindlin-2 = Kindlin-2-CAAX > Kindlin-2- Δ PH-CAAX = Kindlin-

2-QW > Kindlin-2- Δ PH. As demonstrated in our study this trend in cell spreading and FAs formation reflects the ability of kindlin-2-WT and mutants to bind and activate integrins, which is driven by membrane recruitment, membrane free-diffusion, and 3D nanoscale localization in the integrin layer inside FAs.

- We performed additional experiments and quantification to increase the number of independent experiments concerning the enrichment in FAs of kindlin-2 mutants (modified Fig. 6). The results obtained confirmed what we have found in the original manuscript and follow the trend: Kindlin-2 = Kindlin-2-K390A > Kindlin-2- Δ PH-CAAX = Kindlin-2-L357A > Kindlin-2-QW > Kindlin-2- Δ PH > cytosolic mEos2.
- Kindlin-2 dwell-time and rearward motion in mature FAs. Finally, we generated super-resolved time-lapses and kymographs (Fig. 2) to measure kindlin-2 dwell-time in mature FAs and to test whether kindlin-2 is moving rearward inside FAs as demonstrated for talin. First, we found that immobilization durations were much shorter for kindlin-2 compared to talin-1 (Fig. 2h, i). Analysis of longer kymographs showed that the fraction of immobilized kindlin-2 undergoing retrograde flow with speed above 2 nm.s⁻¹ was around 40 % similar to what was measured for talin-1 (Fig. 2j, k)(Rossier et al., 2012). Our results suggest that the interaction of kindlin-2 with stationary integrins is more prevalent than with F-actin, because the fraction of F-actin moving rearward above 2 nm.s⁻¹ is around 75 %(Rossier et al., 2012). We added these new results in Fig. 2, and we mentioned these results in the results section (p.6) and the discussion section to establish a model with the sequential formation of a transient tripartite integrin-kindlin-talin complex (p.12).
- We have performed all the additional quantifications that has been proposed by the reviewers (Fig. 2, 5, 6, 7 and Supplementary Fig. S3, S4, S9).

1) the authors show that beta1 and beta3 integrins seem differently sensitive to impairment of the interaction with kindlin in Figure 1. It is not clear how useful and important these data are for the remaining of the story as they are not discussed further.

We agree with the reviewer, we should have included in the discussion a section about the differences we found between 131- and 133-integrins. This is particularly interesting since different integrin classes are present in the same mature FAs and use distinct mechano-transduction and signaling pathways that cooperate to control FAs structure and functions, such as migration, rigidity sensing and signaling.

In a previous study, we demonstrated a horizontal nano-partitioning and nanoscopic dynamics specific to each integrin α 5131 and α v133 within the FAs; 133-class integrins are stationary and enriched within FAs, whereas 131-class integrins are less enriched and display rearward movements (Rossier et al., NCB 2012). Our results indicated that specific classes of α /13 integrins (α 5131 and α v133) act as distinct 'nanoscale adhesion units' within an individual FA with specific dynamics, organization and force transmission of F-actin motion to the ECM (fibronectin). Thus, nano-scale partitioning of proteins inside could induce functional partitioning that will control the assembly and the mechanical functions of integrin adhesions.

In line with this hypothesis, the Fässler group demonstrated using genetically engineered cells and quantitative proteomics that specific α v- and 131-class integrins use distinct mechano-transduction and signaling pathways that cooperate to control adhesion site assembly, F-actin organization and ECM rigidity sensing (Schiller et al., NCB 2013). Importantly, they showed that α v-class integrins link GEF-H1/RhoA/mDia to stress fiber formation, while 131-class integrins link kindlin-2 and the IPP complex to myosin II activation. These results are in line with

the stronger dependence on kindlin-2 we found for 131-integrin activation ($\alpha 5131$ in our MEFs) compared to the activation of 133-integrin ($\alpha v133$ in our MEFs).

Furthermore, the use of fibronectin micropatterned substrates emphasized the distinct localization of αv - or 131-class integrins in FAs, probably induced by different responses to forces (Schiller et al., NCB 2013). Inhomogeneous localization of αv - or 131-class integrins implies that also their associated signaling proteins are probably segregated. Altogether these findings suggest that nanoscale-partitioning of integrins in FAs reflects a segregation of specific signaling properties and signaling tasks that cooperate to determine the functions of FAs.

In addition, despite the fact that $\alpha v133$ and $\alpha 5131$ are both connecting the ECM (i.e. fibronectin) to actin, their abilities to sense and transmit forces are distinct. Several reports have shown that increased generation of extracellular forces causes strengthening of the $\alpha v133$ integrin/F-actin linkages and thus rigidity sensing, while the strength of the $\alpha 5131$ integrin/F-actin connection seems unaffected by force generation (Giannone, JCB 2003; Choquet Cell 1997; Roca-Cusachs, PNAS 2009). On the contrary, force-dependent conformational transitions re-enforces $\alpha 5131$ integrin binding to FN (Friedland, Science 2009; Kong, JCB 2009). This ability is based on the formation of “catch” bonds, which were not yet reported for $\alpha v133$ integrins. Finally $\alpha 5131$ integrin/ECM bonds generate and resist to higher forces compared to $\alpha v133$ integrin/ECM bonds (Schiller, NCB 2013; Roca-Cusachs, PNAS 2009).

To highlight the differences between 131- and 133-integrins in mature FAs and the potential selectivity of kindlin-2 action on 131-integrins we added the following paragraph in the discussion:

“Integrin activation by kindlin-2 in FAs is integrin selective

Different integrin classes are present in the same mature FAs and use distinct mechano-transduction and signaling pathways that cooperate to control FAs structure and functions, such as migration, rigidity sensing and signaling. In a previous study, we demonstrated a nanoscale horizontal partitioning and specific dynamics for integrin $\alpha 581$ and $\alpha v83$ within FAs; 83-class integrins are stationary and enriched within FAs, whereas 81-class integrins are less enriched and display rearward movements (Rossier NCB 2012). Furthermore, the use of fibronectin micropatterned substrates emphasized the distinct localization of αv - or 81-class integrins in FAs (Schiller et al., NCB 2013). These results indicated that specific classes of $\alpha/8$ integrins ($\alpha 581$ and $\alpha v83$) act as distinct ‘nanoscale adhesion units’ within an individual FA with specific dynamics, organization and force transmission of F-actin motion to the ECM. Altogether these findings suggest that nanoscale-partitioning of integrins in FAs reflects a segregation of specific signaling properties and mechanical tasks that cooperate to determine the functions of FAs. Inhomogeneous localization of αv - or 81-class integrins implies that their associated signaling proteins are also probably segregated. In line with this hypothesis, a proteomic study showed that 81-class integrins link kindlin-2 and the ILK/Pinch/Parvin complex to myosin II activation (Schiller, NCB 2013), indicating that kindlin-2 is preferentially coupled to the functions of 81-integrin rather than 83-integrin. This is consistent with our sptPALM results, which show that the fraction and dwell-time of 81-integrin immobilizations in mature FAs are more sensitive to kindlin-2 binding than for 83-integrin immobilizations. Indeed, we observed that an integrin mutation decreasing interaction with kindlin-2 has more effects on 81-integrins than 83-integrins either inside and outside FAs. Furthermore, Mn^{2+} treatment triggering cell spreading (i.e. early integrin activation) depends on the presence of kindlin(Theodosiou et al., 2016). However, our results revealed that Mn^{2+} treatment is not impaired for 83-759A (kindlin

mutant) but is impaired for 81-795A (kindlin mutant), suggesting again that 81-integrin activation critically relies on kindlin-2. In addition, α v83- and α 581-integrin abilities to sense and transmit forces are distinct. Increased application of extracellular forces causes strengthening of linkages between α v83 integrin and F-actin cytoskeleton and thus rigidity sensing, while the strength of the α 581 integrin/F-actin connection seems unaffected by force generation (Giannone, JCB 2003; Choquet Cell 1997; Roca-Cusachs, PNAS 2009). On the extracellular side, α 581 integrin binding to FN is stabilized in a force-dependent conformational transition (Friedland, Science 2009; Kong, JCB 2009). This could explain why α 581 integrin/ECM bonds generate and resist to higher forces compared to α v83 integrin/ECM bonds (Schiller, NCB 2013; Roca-Cusachs, PNAS 2009). An interesting model could be that the activation of 81-integrin is mainly triggered by biochemical reactions such as kindlin-2 binding while the activation of α v83-integrin could be mainly promoted by force generation via talin.”

2) a number of kindlin-2 (K2) mutants are overexpressed in MEFs for the sptPALM experiments, I assume that with the exception of fig 7 and S7, where MEFs K1 and K2 KO are used, all other MEFs are wild-type, thus expressing kindlin WT. I understand the co-expression of fluorescent K2 mutants in normal MEFs allows to track the behavior of the mutated kindlin in the context of normally formed FAs. However, it would be very useful to relate the data on mobility and 3D location with data of FA features: are the number and morphology of FAs in MEFs overexpressing the K2 mutants (QW, L357A, K390A, DeltaPH) similar to wild-type MEFs? In Fig S8 important parameters are reported in Kind KO MEFs expressing K2-DeltaPH and K2-DeltaPH-CAAX mutants. These parameters should be provided also for the other mutants used. This would show the effect of altered mobility and/or 3D localization on FA properties.

Just as a reminder, in our manuscript, using single protein tracking, super-resolution microscopy and functional assays, we established the link between the function of a critical integrin activator, namely kindlin-2, and its molecular behavior and 3D nanoscale localization. In particular, we showed that kindlin efficient interaction with integrins inside FAs could not result from its cytosolic diffusion alone, but requires kindlin membrane diffusion to drive kindlin in the proper FAs functional layer. Thus our study reveals that the molecular dynamical behavior of a protein in a subcellular compartment is directly linked to its function.

We thank the reviewer to have asked for these additional experiments and quantifications, since these new additional results obtained strengthen the message concerning the link between the function of kindlin-2 and its molecular behavior inside and outside FAs.

The reviewer is correct, at the exception of Fig. 7 and Fig. S8 and S9, we performed experiments in wild-type MEFs transfected with kindlin-2 mutants. The reason for this is, as pointed out by the reviewer, to study the diffusive behavior of kindlin-2 mutants in the context of “normal” FAs. The other reason was to perform experiments in the same MEFs for β 1-integrin and β 3-integrin mutants and kindlin-2 mutants. Note that the MEFs used in the current manuscript, are the same than the one used in Rossier et al., Nature Cell Biology 2012.

In experiments performed in WT MEFs expressing the kindlin-2 mutants (QW, L357A, K390A, Δ PH), we did not observe obvious effects on FAs number and morphology. For example, we observed no obvious increased difficulty in finding MEFs with mature FAs when expressing kindlin-2- Δ PH, which is the kindlin-2 mutant with the largest deficit in FAs formation in rescue experiments using Kindlin-1,2 KO MEFs. This

suggests that the function of endogenous kindlin-1 and kindlin-2 on FAs formation is not altered by dominant negative effects triggered by the expression of exogenous kindlin-2 mutants. Note that the level of expression of transfected kindlin-2 mutants is about 1.5 fold higher than the endogenous kindlin-2 (Supplementary Fig. S4). We did not quantify FAs number or morphology in WT MEFs transfected with kindlin-2 mutants, to avoid measurements that would be biased by the presence of the endogenous kindlin-2 and kindlin-1. Instead, we performed those experiments and quantifications in kindlin-1,2 KO MEFs. To link the diffusive behavior of kindlin-2 wild-type and mutants (kindlin-2-WT, kindlin-2- Δ PH, kindlin-2-QW, kindlin-2-L357A) with their functions, we confirmed that their diffusive behaviors were identical in WT MEFs and kindlin-1,2 KO MEFs (Supplementary Fig. S8). Then we quantified cell spreading, FAs numbers and FAs morphology (Fig. 7 and Supplementary Fig. S9).

In the original version of the manuscript we performed rescue experiments in kindlin-1,2 KO MEFs concerning FAs formation for kindlin-2-WT, kindlin-2- Δ PH, and kindlin-2- Δ PH-CAAX. We found that expression of kindlin-2-WT rescue the formation of FAs. The stronger impairment in rescuing FAs formation was found for kindlin-2- Δ PH, which displays impaired membrane free diffusion. Supporting our hypothesis that kindlin-2 membrane diffusion is crucial for kindlin-2 FAs recruitment and functions in FAs, we found a partial rescue of FAs formation by restoring membrane free-diffusion with kindlin-2- Δ PH-CAAX.

We agree with the reviewer that experiments on FAs formation using kindlin-2-QW and kindlin-2-CAAX should have been included in the original version of the manuscript, to mirror the experiments performed on cell spreading: kindlin-2, kindlin-2-QW, kindlin-2- Δ PH, kindlin-2- Δ PH-CAAX and kindlin-2-CAAX.

In the revised manuscript we performed those additional experiments on FAs formations and the associated quantifications for kindlin-2-QW and kindlin-2-CAAX.

As shown in the original version of the manuscript, kindlin-2-QW displayed a decreased immobile fraction and increased free-diffusive fraction in FAs as compared to kindlin-2-WT (Fig. 3b,d), and kindlin-2-QW enrichment in FAs was decreased compared to kindlin-2-WT (Fig. 6b, f). Thus, it was important to include this mutant in the analysis of FAs formation, to link its diffusive behavior to its function in FAs. As for kindlin-2-CAAX, it is a control which shows that adding a CAAX membrane anchor to WT kindlin-2 do not trigger an enhanced activity of kindlin-2-WT during cell spreading. Using the same CAAX membrane anchor to restore kindlin-2- Δ PH membrane free-diffusion and localization in the integrin layer increases its recruitment to FAs, and its function during cell spreading and FAs formation. However, we did not perform these experiments with kindlin-2-L357A and kindlin-2-K390A which have no obvious effects on kindlin-2 diffusive behavior (kindlin-2-L357A: Supplementary Fig. S4 (MEFs) and S8 (KindKO cells); kindlin-2-K390A: Supplementary Fig. S5), membrane recruitment (Fig. 4f) and FAs enrichment (Fig. 6) compared to kindlin-2- Δ PH and kindlin-2-QW.

Our new results show that expression of kindlin-2-QW rescue the formation of FAs at the same level as kindlin-2- Δ PH-CAAX (Fig. 7 and Supplementary Fig. S9). These results confirm that the kindlin-2-QW mutant can still bind integrins (as found in sptPALM experiments; Fig. 3, as discussed in the manuscript) but also show that kindlin-2-QW mutant could also partially activate integrins. On its hand, kindlin-2-CAAX rescue the formation of FAs to the same level as kindlin-2, demonstrating that the addition of a CAAX membrane anchor to WT kindlin-2 do not trigger an enhanced activity of kindlin-2-WT during FA formation (Fig. 7 and Supplementary Fig. S9).

Importantly, these new results on FAs formation (Kindlin-2 = Kindlin-2-CAAX > Kindlin-2- Δ PH-CAAX = Kindlin-2-QW > Kindlin-2- Δ PH) follow the trend quantified for these mutants concerning FAs enrichment (Fig. 6), and cell spreading (Fig. 7) and probably reflects the ability of kindlin-2 WT and mutants to bind and activate integrins.

3) could the authors explain why K2-DeltaPH is immobile outside FAs?

Deletion of the entire PH domain of kindlin-2 (kindlin-2- Δ PH) strongly inhibited membrane free diffusion, both inside and outside FAs (Fig. 4a,c,d, Supplementary Movie S2), confining most of kindlin-2- Δ PH free-diffusion into the cytosol (Fig. 4f). A protein freely diffusing in the cytosol is characterized by coefficient of diffusion (D) around 100 times faster than the one of a protein freely diffusing on the membrane. With our acquisition parameters (50 Hz) we are not able to reconnect trajectories of proteins freely diffusing in the cytosol with very fast D. Thus with our method we are not able to reconnect trajectories of a protein freely diffusing in the cytosol and thus to measure its fast D.

Instead, our method allows us to record the trajectories of a cytosolic protein that freely diffuses on the membrane if this cytosolic protein interacts with membrane components that display free-diffusion (for instance, kindlin-2 (Fig. 2), PH-domain (Fig. 4), CAAX (Rossier et al., NCB 2012), kindlin-2-CAAX (Fig. S7)). Similarly, our method allows us to reconnect the trajectories of a cytosolic protein if this protein is associated with immobilized membrane components, such as proteins binding to immobile integrins inside and outside FAs, like for instance paxillin (Fig. S6) or talin (Fig. 2).

In the case of kindlin-2- Δ PH, the fraction of kindlin-2- Δ PH freely diffusing in the cytosol with very fast D is missing in the distribution of D both inside and outside FAs for the reason mentioned above. However, since kindlin-2- Δ PH lacks membrane free-diffusion, kindlin-2- Δ PH interacting transiently with immobile binding partners outside or inside FAs represents the largest fraction at the membrane level. This explains why kindlin-2- Δ PH appears mainly immobile outside FAs (Fig. 4).

We demonstrated that the PH domain of kindlin-2 is not only driving kindlin-2 membrane recruitment and free-diffusion (Fig. 4), but is also leading to the subsequent interaction with integrins (Fig. 6, 7). Perhaps the immobile fraction of kindlin-2- Δ PH corresponds to residual interactions with integrins occurring both inside and outside FAs. Alternatively, kindlin-2- Δ PH could bind to other proteins, actin, ILK or still unknown binding partners (Bledzka JCB 2016; Huet-Calderwood JCS 2014).

To clarify this point, we added in the method section a paragraph explaining this bias of the SPT method, page 17:

“With the acquisition frequency and TIRF illumination used in our experiments, it is impossible to reconnect trajectories lasting at least 260 ms (≥ 13 points) for a protein freely diffusing in 3D within the cytosol. Therefore, the very low occurrence of free-diffusing trajectories inside or outside FAs for kindlin-2- Δ PH and paxillin indicates that these proteins are mostly moving in the cytosol.”

Kindlin-2 and kindlin-2- Δ PH displays similar immobile fractions inside FAs (Fig. 4) but the density of immobilization is clearly decreased for kindlin-2- Δ PH compared to kindlin-2 (Fig. 2a vs Fig. 4a) reflecting a decreased enrichment in FAs for kindlin-2- Δ PH (Fig. 6f). Outside FAs, since kindlin-2- Δ PH is mainly freely diffusing in the cytosol, the dominant apparent diffusive behavior is immobilization as explained above, but the density of immobilization outside FAs is very low.

This answer is also related to the question asked in the point#5 (see below).

4) data in Fig 1e,f are also shown in Fig S1 e,f: is this necessary?

We added these distributions of coefficient of diffusion again in the Supplementary Fig. S1, to help appreciate the differential effects of Mn^{2+} treatment on 131-integrin and 133-integrin activation. These data could be removed if the reviewer thinks that they are not necessary.

5) why is beta3 wild-type integrin immobile outside FAs (Fig 1h)? Why is talin so highly immobile outside FAs (Fig 2d)? Why is 70% of paxillin immobile outside FAs (Fig S5c)?

Those 3 queries are all related to the point #3.

First concerning the significant fraction of 133-integrin immobilization outside FAs (Fig. 1h).

Immobilizations are occurring outside FAs for most proteins of integrin-dependent adhesions. This includes 133-integrin, 131-integrin, paxillin, talin, kindlin. However, there are well-defined differences between immobilizations occurring inside versus outside mature FAs:

- The density of immobilization in FAs is much higher than outside FAs. This is discussed below, and is valid at least for 133-integrin, 131-integrin, talin, kindlin, paxillin.
- The dwell-time of immobilizations are shorter outside than inside FAs. This was quantified in the article Rossier et al., Nature Cell Biology 2012 for 133-integrin (Fig. 3c NCB 2012), 131-integrin (Fig. 5e NCB 2012) and talin (Fig. 4h NCB 2012).
- Immobilizations outside FAs for 133-integrin and 131-integrin are decreased when using integrin mutants with decreased interactions with fibronectin (133-integrin-D119Y (Fig. 2-3 NCB 2012), 131-integrin-D130Y (Fig. 5 NCB 2012)) or with intracellular binding partners triggering integrin activations (talin (131-Y783A or 133-Y747A in Fig. 1), kindlin (131-Y795A or 133-Y759A in Fig. 1)). Reciprocally, immobilizations outside FAs for kindlin-2 are decreasing when using a kindlin-2 mutant with impaired binding to integrins (kindlin-2-QW in Fig. 3).
- Immobilizations outside FAs are lower for control proteins as mEOS2 fused to the trans-membrane domain of the PDGF receptor (Fig. 1 NCB 2012) or anchored to inner leaflet lipids CAAX-mEOS2 (Fig. 1 NCB 2012).

Altogether, these results indicate that these immobilizations outside FAs are associated with specific integrin immobilizations and not aspecific immobilizations. They might be triggered by transient activation of integrins occurring outside mature FAs that are required for the initiation of early integrin adhesions, leading to the formation of nascent adhesions (NAs). Actually, reviewer#1 asked us to perform sptPALM experiments on kindlin-2 during the formation of early integrin-adhesions, which is occurring in the lamellipodium. We performed these experiments with kindlin-2-WT and kindlin-2-QW with decreased binding to integrins. These results are presented in Supplementary Fig. S3. They show a significant fraction of immobilized kindlin-2, inside and outside the lamellipodium. Again kindlin immobilizations are also decreased by mutations which reduce binding to integrin. Thus transient immobilizations of integrin adhesion proteins outside mature FAs could correspond to transient activation of integrins, reflecting a basal level of integrin activation. These transient activations could lead to initiation of nascent adhesion in the lamellipodium.

Second, concerning the large fraction of immobilization for talin and paxillin outside FAs:

We want to emphasize that the immobilization fraction is not always correlated with the immobilization density or recruitment within a subcellular structure. For example, it is possible to obtain similar immobilization fractions but with very different immobilization densities. For the same area, it is possible to obtain 10 stationary and 10 free-diffusive trajectories, or 100 stationary and 100 free-diffusive trajectories, the immobilization fraction (50%) will be the same but the density will be an order of magnitude different.

As noticed by the reviewer, this is the case for talin-1, which display similar levels of immobilization inside versus outside FAs (82% vs 73%, Fig. 2), but with much higher density of immobilization inside versus outside FAs (~ 5 time more detections inside versus outside FAs, Fig. 4f in Rossier et al., Nature Cell Biology 2012). This is also the case for paxillin which possesses large immobile fractions inside and outside FAs but is specifically enriched in FAs (Supplementary Fig. S6). Related to the first part of the question, we think that paxillin and talin immobilizations outside mature FAs could be associated with transient activation of integrins that could occur outside mature FAs, as a basal level of integrin activation that could lead to the initiation of nascent adhesions.

There is a clear difference in SPT experiments between proteins that are freely diffusing on the plasma membrane (transmembrane proteins, and cytosolic proteins which interact with diffusive components of the membrane): integrins, kindlins, CAAX-tagged proteins, Rho GTPases (Mehdi Current Biology 2019); and cytosolic proteins that do not associate with diffusive components of the membrane: talin, paxillin, kindlin-2- Δ PH...

For proteins freely diffusing on the plasma membrane, a decreased immobilization fraction in conjunction with an increased fraction of membrane free-diffusion reflects a transfer in population from an immobile pool engaged with binding partners to a freely diffusing pool detached from this binding partner. Thus, there is a transfer between immobile and freely diffusive fractions in conjunction with a decreased density of immobilization. This is the case in FAs, for instance for kindlin-2 versus kindlin-2-QW (Fig. 3 in MEFs and Supplementary Fig. S8 in Kind^{Ko} cell), or β 1-integrin versus β 1-integrin-Y795A (Fig. 1).

For cytosolic proteins that do not associate with diffusive components of the membrane, a decreased immobilization in a specific sub-cellular compartment such as FAs, mainly lead to a decreased density of immobilization. This does not automatically change the fraction of immobilization, since there is no transfer from immobilization to free diffusion on the membrane, but transfer to free-diffusion in the cytosol that as explained earlier cannot be detected in SPT experiments.

6) the Diff coefficients are calculated from the small fraction that is mobile, which is very small inside FAs, at the same time the changes in D values are also very very modest: they are statistically different, but what do these tiny changes tell biologically? Could the authors strengthen their interpretation of these data?

Indeed, the coefficient of diffusion (D) for free-diffusive events (D_{diff}) are calculated from the fraction of freely diffusive molecules. These fractions could be small especially in mature FAs.

First, we want to emphasize that in most cases there are huge differences in D_{diff} when compared between inside and outside FAs. Although, changes in D_{diff} in mature FAs might be small when comparing

different conditions, they are statistically different as pointed out by the reviewer. For instance, the D_{diff} in FAs for $\beta 3$ -integrin and $\beta 1$ -integrin are significantly slower compared to integrin mutants defective in binding to kindlin/talin ($\beta 3$ -integrin-Y759A/ $\beta 3$ -integrin-Y747A; $\beta 1$ -integrin-Y795A/ $\beta 1$ -integrin-Y783A) (Supplementary Fig. S1). We found similar results in a previous study for integrin mutants defective in binding to fibronectin/talin (Rossier et al., NCB 2012: Fig. 2g). Similarly, the D_{diff} in FAs for kindlin-2 is slightly but significantly slower compared to the kindlin-2-QW mutant defective in binding to integrins (Fig. 3e). Similarly, the D_{diff} in FAs for kindlin-2 is clearly slower compared to kindlin- Δ PH-CAAX (Supplementary Fig. S7d), in line with the faster D_{diff} found outside FAs for kindlin- Δ PH-CAAX compared to kindlin-2 (Supplementary Fig. S7d).

Importantly, we also found significant differences in the D_{diff} inside and outside FAs for integrins and kindlins (Supplementary Fig. S1 and Fig. 2), D_{diff} for kindlin being slower than for integrins (Fig. 2). These results suggest that kindlin-2 and integrins are not diffusing at the plasma membrane when bound to each other. This is consistent with a model where kindlin-2 is entering FAs without being co-associated with integrins outside FAs, moving within FAs using membrane free diffusion to reach integrins and trigger their immobilization inside FAs.

We think that D_{diff} correspond to the genuine D of free-diffusion for a specific protein, but also conceal transient interactions of this protein with surrounding binding partners that could not be captured because of limited acquisition frequencies inherent to any experiments. We explain this below:

Since it is impossible to obtain an infinitely fast acquisition frequency, we are most of the time in SPT quantifying apparent coefficient of free-diffusion (D) and not absolute D . If we take integrin as an example: in principle integrin-WT in its activated or inactivated states, when not interacting with other proteins (fibronectin, talin, kindlin) or specific membrane domains, should diffuse at the same speed (identical D). In other words, the diffusive behavior will be determined by the physical parameters of the protein and the environment in which the protein is evolving (hydrodynamic radius of the protein, viscous drag of the cytosol and the membrane, collision within the membrane...).

Now, if we introduce specific binding events in between periods of free-diffusion, but if those binding events are most of the time lasting less than what could be temporally resolved in the experimental and analysis framework, the studied protein will still be identified as a freely diffusive but will be characterized by a decreased rate of free-diffusion, i.e. D_{app} will be inferior compared to D absolute. The Mean Squared Displacement (MSD) will remain linear but with a lower slope. The disparity in the D_{diff} values between integrin-WT and integrin- $\beta 3$ -D119Y (Fig. 2 NCB 2012) or integrin- $\beta 3$ -Y747A (Fig. 2 NCB 2012, or Supplementary Fig. S1 in the present manuscript) suggests that integrin-WT undergoes additional transient interactions.

In our experimental conditions, since we use a minimum of 10 points of the MSD to analysis the diffusive behavior of proteins, we are able to capture immobilization events lasting more than 200 ms at 50 Hz (10 x 20 ms) (see methods). In future studies, it will be interesting to increase the frequency of acquisitions of sptPALM experiments to investigate these much transient interactions between integrins and regulators. For instance, we could perform sptPALM experiments at 333 Hz, enabling to capture immobilization events of 30 ms, as we did in a study focused on Rac1 immobilizations at the tip of the lamellipodium of migrating cells (Mehidi et al., Current Biology 2019).

7) Figure 6: I appreciate the complexity of the imaging approaches used in this study but presenting data in a main figure that are obtained by two or one experiment is not sufficiently sound. In Fig. S6, four experiments are mentioned in which the same mutants used for figure 6 were used. I therefore assume the authors have the possibility to add more experimental data and strengthen the results shown in Fig. 6. In addition, I am wondering why the information provided in Fig. 6 is actually not shown and discussed earlier in the manuscript, right before the authors embark on sptPALM experiments? What is the reason for not putting these data as supplementary figure?

Following the reviewer advice, we performed additional experiments and quantification to increase the number of independent experiments to at least 3. Displayed results now correspond to pooled data from at least 3 independent experiments (kindlin-2-WT: 36 cells, kindlin-2-QW: 24 cells, kindlin-2- Δ PH: 27 cells, kindlin-2- Δ PH-CAAX: 50 cells, kindlin-2-L357A: 34 cells, kindlin-2-K390A: 30 cells, mEos2: 38 cells).

From point #8: Organizing the manuscript as proposed by the reviewer (functional effects on the formation of mature FAs followed by a molecular understanding of what drive these differences) is an interesting alternative to what we chose in the original version of the manuscript (mechanistic to functional). However, we think that both organizations are valid, and we would like to keep the original organization of the manuscript.

Moreover we think that the data displayed in Fig. 6 constitutes a strong argument for the role of membrane diffusion on the function of kindlin-2 as it establishes a clear correlation between kindlin-2 membrane recruitment and accumulation inside FAs.

8) the data provided in Figure 7 are used a conclusive statement to link the biophysical parameters to integrin activation (cell spreading). However, these data are not entirely novel as the eLife paper of the Faessler group already provided the same knowledge. I am therefore wondering: why not using these data as starting point? As motivation to better understand mechanistically what drives these differences? Mobility and nanoscale 3D localization of kindlin would then be the mechanistic explanation. In Fig S8, interesting data are shown for individual FAs formed after expression of two out of the five kindlin-2 mutants used. I think the same parameters should be provided for the QW, L357A and K390A mutants, to link the effects of these mutations to individual FA properties and eventually to cell spreading. Unfortunately, Fig S8 seems not mentioned in the Results section.

The 2016 eLife article from the Fässler group “Theodosiou et al., Kindlin-2 cooperates with talin to activate integrins and induces cell spreading by directly binding paxillin”, is focused mainly on the ability of kindlin to activate integrins during cell spreading and thus the formation of early adhesive structures including Nascent Adhesions (NAs). In contrast, in our current manuscript, we are studying the function of kindlin during integrin activation in mature FAs. Nevertheless, we agree with the reviewer that part of our results concerning cell spreading in Kindlin1,2 KO MEFs rescued with expression of wild-type or mutant forms of kindlin-2 may overlap with some results obtained in this eLife article: this mainly concern experiments performed using kindlin-2-WT and kindlin-2- Δ PH (Fig. 4F in Theodosiou et al.).

However, the experiments performed and the results found are not exactly the same. In the eLife article the authors focused on cells displaying isotropic spreading rapidly after plating (10-30 min.). This isotropic spreading behavior is mainly triggered in Talin-KO fibroblasts after Mn^{2+} integrin activation. In the case of isotropic spreading, the shape of cells is not dependent on the formation of mature FAs but depends on

the formation of early NAs. While in our article we quantified the morphology of cells 4-5 hours after spreading on fibronectin, thus we did not study or quantified isotropic cell spreading. In our case, the final shape of cells depends mainly on the formation and localization of mature FAs in cells. Note that we are fully aware about cell behaviors during isotropic or anisotropic spreading and the ensuing cell polarization, as we characterized and studied these phenomenon in previous articles (Giannone et al., Cell 2004, Cell 2007; Dubin-Thaler et al., Biophysical J 2004).

Note also that we are using the same kindlin-1,2 KO MEFs than in the eLife article, since Reinhard Fässler and Ralph Böttcher are co-authors on our current manuscript. In the eLife study they found that expression of kindlin-2- Δ PH in kindlin-1,2 KO MEFs can rescued isotropic cell spreading but cannot recruit paxillin to NAs. In the results section taken from the eLife paper, p. 10: *“The experiments revealed that the expression of K2-GFP in Kind Ko cells rescued spreading and induced robust paxillin recruitment to 81 integrin-positive NAs (Fig. 4F, G). In contrast, expression of K2- Δ PH-GFP failed to recruit paxillin to 81 integrin-positive adhesion sites at the rim of membrane protrusions (Fig. 4F,G) and induce normal cell spreading (Figure 4— figure supplement 4A) despite proper, although weaker, localisation to 81 integrin-positive adhesion sites (Figure 4—figure supplement 4B,C).”*

However, in our results cell morphology is quantified at later time points, where mature FAs also participate in the final shape of spread cells. In that case kindlin-2- Δ PH is poorly restoring the formation of mature FAs and consequently cell spreading (Fig. 7). These results are thoroughly quantified in the current manuscript. Furthermore, this idea is reinforced by the fact that the effects of different kindlin-2 mutants on the formation of mature FAs (Kindlin-2 = Kindlin-2-CAAX > Kindlin-2- Δ PH-CAAX = Kindlin-2-QW > Kindlin-2- Δ PH) follow the trend quantified for these mutants for cell spreading (Fig. 7b) and mature FAs enrichment, which probably reflects the ability of kindlin-2 WT and mutants to bind and activate integrins in mature FAs. To further demonstrate the correlation between the number of mature FAs and cell spreading area, we plotted a graph displaying the relationship between cell spreading area as function of the number of mature FAs (graph displayed below). (see also answer point #2)

Figure for reviewer #3: Plot of the cell total area (data displayed in Fig. 7e) in function of the number of adhesions developed at the cell surface (data displayed in Fig. 7d) of Kind^{KO} cells (4h after seeding on fibronectin) re-expressing for 2 days mEos2-kindlin-2-WT (light blue), mEos2-kindlin-2- Δ PH (orange), mEos2-kindlin-2- Δ PH-CAAX (green), mEos2-kindlin-2-QW614/615AA (red) or mEos2-kindlin-2-WT-CAAX (dark blue). Each point in the distribution represents the value obtained from a single cell. FAs were drawn manually and cell boundaries were determined by manually setting a threshold on the pixel intensity values using the TI RF GFP-paxillin images as displayed in Fig 7c. GFP-paxillin + mEos2-kindlin-2-WT: 84 cells; GFP-paxillin + mEos2-kindlin-2- Δ PH: 74

cells; GFP-paxillin + mEos2-kindlin-2- Δ PH-CAAX: 62 cells; GFP-paxillin + mEos2-kindlin-2-QW614/615AA: 30 cells; GFP-paxillin + mEos2-kindlin-2-WT-CAAX: 63 cells. The results correspond to pooled data from three independent experiments with at least 8 cells per experiment.

Again, in the eLife study, the focus was not the functions of kindlin during integrin activation in mature FAs. Indeed, this article did not contain quantifications of mature FAs numbers, FAs sizes. Furthermore, no attempt was made to relate FAs numbers and sizes to the final area of spread and polarized cells. Since the current manuscript is focusing on mature FAs, we performed these quantifications for almost all the kindlin-2 mutants described in the manuscript: Kindlin-2-WT, Kindlin-2- Δ PH, Kindlin-2- Δ PH-CAAX, Kindlin-2-QW, and Kindlin-2-CAAX. The eLife article only briefly mentioned the effects of the K2- Δ PH-GFP mutant on formation of mature FAs, but without quantification since this article was focusing on early NAs in the lamellipodium. In the results section, p. 10: *“Interestingly, mature FAs in K2- Δ PH-GFP-expressing cells were prominent after 30 min and contained significant amounts of paxillin, indicating that paxillin is recruited to mature FAs in a kindlin-2-independent manner (Figure 4F). These findings indicate that the PH domain of kindlin-2 directly binds the LIM3 domain of paxillin and recruits paxillin into NAs but not into mature FAs.”* Therefore, for all the reasons explained above, we think that the results of Fig. 7 are distinct from but complementary to the results obtained in the previous eLife article.

As explained in the point #7, organizing the manuscript as proposed by the reviewer #3 (functional effects on the formation of mature FAs followed by a molecular understanding of what drive these differences) is an interesting alternative to what we chose in the original version of the manuscript (mechanistic to functional). However, we think that both organizations are valid, and we would like to keep the original organization of the manuscript as we are providing new evidences of the role of kindlin-2 on mature FAs formation.

Concerning the results of Fig. 7 and Supplementary Fig. S9, focusing on mature FAs formation in Kindlin1,2 KO MEFs expressing various kindlin-2 mutants, we have already addressed this concern in point #2, which includes a question similar to that in point #8.

Point#2: “In Fig S8 important parameters are reported in Kind KO MEFs expressing K2-DeltaPH and K2-DeltaPH-CAAX mutants. These parameters should be provided also for the other mutants used. This would show the effect of altered mobility and/or 3D localization on FA properties.”

Point#8: “In Fig S8, interesting data are shown for individual FAs formed after expression of two out of the five kindlin-2 mutants used. I think the same parameters should be provided for the QW, L357A and K390A mutants, to link the effects of these mutations to individual FA properties and eventually to cell spreading. Unfortunately, Fig S8 seems not mentioned in the Results section.”

In summary as explained in point#2. In the original version of the manuscript we performed these experiments and analysis for kindlin-2-WT, kindlin-2- Δ PH, and kindlin-2- Δ PH-CAAX. In the revised version of the manuscript we performed new experiments and analysis for kindlin-2-QW and kindlin-2-CAAX. However, we did not perform these experiments with kindlin-2-L357A and kindlin-2-K390A which have no obvious effects on kindlin-2 diffusive behavior (kindlin-2-L357A: Supplementary Fig. S4 (MEFs) and S8 (KindKO cells); kindlin-2-K390A: Supplementary Fig. S5), membrane recruitment (Fig. 4f) and FAs enrichment (Fig. 6) compared to kindlin-2- Δ PH and kindlin-2-QW. We invite the reviewer to refer to the answer in point #2, which is fully developed and detailed earlier.

Reviewer #1 (Remarks to the Author):

The authors answered all my questions exhaustively, clarifying all the puzzling points of the original version. I now recommend its publication in Nature Communications.

Renaud Poincloux

Reviewer #2 (Remarks to the Author):

In this revision, the authors have performed a comprehensive list of additional experiments, and clarified in details the issue raised for the initial submission. I am satisfied with the revision and support its acceptance.

Reviewer #3 (Remarks to the Author):

The revised manuscript by Orré and colleagues addresses all my concerns and the explanations in the rebuttal are very detailed and convincing. I fully support publication of the revised version as it is presented here.